# Enhancing coastal winds and surface ocean currents with deep learning for short-term wave forecasting

Manuel García-León[1], José María García-Valdecasas[1], Lotfi Aouf[2], Alice Dalphinet[2], Juan Asensio[1], Stefania A Ciliberti[1], Breogán Gómez[1], Víctor Aquino[1], Roland Aznar[1], Marcos Sotillo[1]

[1]Nologin Oceanic Weather Systems SLU, Santiago de Compostela, 15705, Spain
[2]Meteo-France, Departement Marine et Oceanographie, Toulouse, 31100, France

*Correspondence to*: Manuel García-León (manuel.garcia@nowsystems.eu)

**Abstract.**

Accurate short-term wave forecasts are crucial for numerous maritime activities. Wind and surface currents, the primary forcings for spectral wave models, directly influence forecast accuracy. While remote sensing technologies like Satellite Synthetic Aperture Radar (SAR) and High Frequency Radar (HFR) provide high-resolution spatio-temporal data, their integration into operational ocean forecasting remains challenging. This contribution proposes a methodology for improving these operational forcings by correcting them with Artificial Neural Networks (ANNs). These ANNs leverage remote sensing data as targets, learning complex spatial patterns from the existing forcing fields used as predictors. The methodology has been tested at three pilot sites in the Iberian-Biscay-Ireland region: (i) Galicia, (ii) Tarragona and (iii) Gran Canaria.

Using SAR as a reference, the ANN corrected winds present Root Mean Square Deviation (RMSD) reductions close to 35% respect to ECMWF-IFS, and improvements close to 3% for the scatter-index. Surface currents are also improved with ANNs, reaching speed and directional biases close to 2 cm/s and 6º and correlation close to 35% and 50%, respectively. Using these ANN forcings in a regional spectral wave model (Copernicus Marine IBI-WAV NRT) leads to improvements in the Wave Height ($H_{m0}$) bias and RMSD around 10% and 5% at the NE Atlantic. Mean wave period ($T_{m02}$) also improves, with reductions of 17% and 5% in bias and RMSD. Preliminary moderate improvements were also present in extreme events (e.g. storm Arwen at Galicia, November 2021), as the $H_{m0}$ was corrected close to 0.5m and $T_{m02}$ by around 0.4s. However, properly quantifying this impact requires further assessment.

## 1 Introduction

Short-term wave forecasting is gaining momentum as a reliable source for decision making and daily operational planning at the coast. Proper characterization of surface waves is relevant in many sectors, namely (i) shipping, (ii) offshore and coastal structures, (iii) marine renewable energy, (iv) disaster risk mitigation, (v) coastal management and (vi) recreational activities (Giunta et al., 2024, El Serafy et al., 2023).

Third-generation spectral wave models are suitable for the regional scale because they address wave generation, propagation, and non-linear interactions (WISE group, 2007). They are computationally affordable and reliable for many applications (Capet et al., 2020). However, errors tend to increase at the coast, due to the compound effect of many factors: (i) accuracy in the forcings; (ii) limitations in the physical parameterizations; (iii) overlooked physical processes (e.g. river discharge, coastal morphodynamics, wave-biota interaction, wet-and-dry), (iv) model discretization and numerical schemes, etc.

Operational services are applying diverse strategies to cope with these errors: (i) data assimilation (Aouf et al., 2006, Aouf et al., 2015), (ii) reduce biases and errors in the inputs (Durrant et al., 2013, Silva et al., 2022, Zieger, 2025), (iii) implement latest research in coastal wave processes as wave-growth (Breivik et al., 2022, Cavalieri et al., 2024a), wave-current interaction (Staneva et al. 2015, Kanarik et al. 2021), wave breaking (Salmon et al., 2015), bottom friction (Zijlema et al., 2012) and triad interaction (Salmon et al., 2016), (iv) increase the wave model resolution (Sánchez-Arcilla et al., 2014).

Reducing biases and errors in the forcings has certain advantages over other strategies, because it does not increase computational time by changing the model resolution. Lower forcing errors also bounds a source of uncertainty, assisting in the implementation of the other two strategies. Main inputs in the spectral wave models are (i) wind fields, (ii) surface ocean currents and (iii) bathymetry. At regional scale models, the bathymetry is considered as a static forcing, because the rate of change for short-term forecasting (i.e. a few days) is only significant at shallow waters. Hence, for the forecast window considered, winds and currents can be considered as the main dynamic forcings.

Errors in wind fields may be one of the causes for low performance in wave forecasts. Third-generation spectral wave models exhibit sensitivity to changes in the wind forcings (WISE group, 2007, Cavaleri et al., 2018). Despite that the accuracy of operational atmospheric products as ECMWF-IFS is good enough, due to data assimilation and physics parameterizations, they still face systematic biases (Hersbach et al., 2010, Sandu et al., 2013, Sandu et al., 2020). Most of these biases are due to the physics, and limitations imposed by the coarse resolution of the atmospheric forcings, usual in short-term forecasting (Belmonte and Stoffelen, 2019, Cavalieri et al., 2024b).

A strategy for reducing these systematic wind biases has been the combination of model outputs with remote sensing data, especially with scatterometer data (Hauser et al., 2023). Relevant derived products include Copernicus Marine WIND-TAC (e.g. WIND_GLO_PHY_L4_MY_012_006, Giese and Stoffelen (2024)) and ERAstar (Portabella et al., 2022). At the coastal zone, though, scatterometer measurements may be heavily affected by land contamination. But also note that this limitation could be reduced in the near future with new correction schemes (Grieco et al., 2024).

Errors in local coastal winds, penalize the wind-sea fraction of the wave spectra (Donelan et al., 1992). Wind products derived from Synthetic Aperture Radar (SAR) data from ESA Sentinel-1 mission (Torres et al., 2012), are complementing this gap at the coastal zone. SAR high spatial resolution is relevant for the coastal zone, as it allows to capture fine-scale wind structures, such as coastal jets, katabatic winds, and land-sea breezes, that are often unresolved by the relatively coarser resolution of numerical weather prediction models (Mouche et al., 2019, Wei et al., 2020, Stopa et al., 2022).

Another main forcing at the coastal zone are the surface currents. Including currents in spectral wave models can reduce the errors on significant wave heights by more than 30% in some macrotidal environments, such as at the coast of Brittany

(France) (Ardhuin et al., 2012). Wave-current interaction can affect (i) refraction due to currents, (ii) shoaling, and (iii) current-driven frequency shifting (Staneva et al., 2017; Cavalieri et al., 2018; Law Chune and Aouf, 2018; Bruciaferri et al., 2021; Calvino et al., 2022). While numerical wave models have demonstrated considerable skill in predicting wave conditions, the predictive skill of coastal circulation models remains comparatively lower, particularly in complex coastal regions (Fringer et al., 2019, García-León et al., 2022).

High-Frequency (HF) radar can remotely measure ocean surface currents by analyzing the Doppler shift of radio waves backscattered by surface waves, a process known as Bragg scattering (Barrick, 1972). Combining radial velocity measurements from multiple sites allows for the derivation of two-dimensional surface current vector fields (Gurgel et al., 1999). These spatio-temporal current fields provide comprehensive information across a range of temporal scales, from short-period fluctuations driven by tides and wind to longer-term variations associated with mesoscale eddies and coastal currents (Chapman et al., 1997).

Consequently, to enhance coastal circulation skill, HF Radar is often integrated with circulation models. For instance, an early attempt was made by Breivik and Sætra (2001), who used data assimilation to increase the accuracy of their simulated current fields in the Skagerrak-Kattegat area.The results were that lower frequencies with periods of more than 6 hours were improved by assimilating HF radar. Later, Stanev et al. (2015) proposed a variational approach to assimilate radial current components from two HF radars into a forecast system at the German Bight (Stanev et al., 2011), based on the General Estuarine Transport Model (GETM, Burchard and Bolding, 2002). The results shown a substantial improvement in tidal prediction accuracy (specially the dominant M2 tidal constituent) and subtidal variability, indicating potential applications for short-term predictions. An extensive review of blending HF-Radar with models can be found in Updyke (2022).

As the observational volume grows and the predictions models become more complex, there are needed data analysis techniques that should allow (i) compression of the information, but (ii) preserving its strong variability. In the last decade, the applicability of Machine Learning (ML) at Geosciences has experienced a exponential growth, and there are available a wide variety of analysis techniques (see De Burgh-Day and Leeuwenburg, 2023 for a review). Artificial Neural Networks (ANNs) (McCulloch et al., 1943, Hornik et al., 1989) are one of the most suitable ML techniques for addressing this challenge because they can model inherent non-linear processes and effectively learn from complex, high-dimensional datasets. Furthermore, ANNs are highly suitable for analyzing dynamic and data-sparse met-ocean variables because they offer superior generalization capabilities, adapt to evolving conditions, and do not require prior assumptions about data distribution. Depending on the problem and the data features, ANNs require varying network structures to ensure effective learning and accurate predictions. Many recent publications have addressed the problem of predicting coastal waves by training ANNs with wave model outputs (Jing et al., 2022, Minuzzi and Farina, 2024), remote sensing (Quach et al., 2020, Atteia et al., 2022) or buoys (Deo et al., 2001, Yang et al., 2021). Also, there have been many proposed ANN models for wind forecasting (see reviews by Wang et al., 2021 and Wu et al., 2022) and surface currents (e.g. Ren et al., 2018, Bradbury and Conley, 2021). Many of these references relies on ANN architectures based on (i) Recurrent Neural Networks (RNNs), when they aim to predict temporal data; or (ii) Convolutional Neural Networks (CNNs) for dealing with spatial data.

This paper aims to propose ANNs for correcting the forcings (winds and surface currents) of spectral wave models. The main hypothesis will be: (i) to use SAR data as the source for correcting wind fields, (ii) to use HF-radar as the source for correcting surface currents, (iii) the correction procedure involves deep learning techniques (i.e. GAN for winds fields and Autoencoders for surface currents), (iv) that the enhanced forcings can be applied to short-term forecasting.

The proposed methodology will be tested in the Iberian-Biscay-Ireland region (IBI) region. The IBI area has been selected due to the availability of the remote sensing data and HF-Radar. The ANN forcings will be tested for improving the accuracy of the Copernicus Marine Service regional wave prediction system (i.e. CMS IBI-WAV NRT). These developments have been conducted in the Copernicus Marine Service Evolution KAILANI (2022 – 2024).

The paper is organized as follows: Section 2 describes (i) the current set-up of the IBI-WAV forecast service and the proposed improvements, (ii) the proposed ANNs architectures for winds and surface currents, (iii) the pilot zones in the IBI region, (iv) the benchmark period for the methodology testing. Section 3 shows (i) the results of each ANN and (ii) the testing of the benchmarks in IBI-WAV. Section 4 discusses the results and its implications; whilst Section 5 concludes this article.

## 2 Material and Methods

This Section will describe the CMS IBI-WAV-NRT service and the general structure of the proposed methodology (termed KAILANI hereafter) (Subsection 2.1). Next, there will be presented the different ANNs architectures (Subsection 2.2 and 2.3), the pilot zones & benchmarks (Subsection 2.4), the list of experiments for assessing the feasibility of KAILANI (Subsection 2.5) and the error metrics that would be used (Subsection 2.6).

### 2.1. The CMS IBI-WAV-NRT service and proposed AI forcing enhancements

Wave analyses and forecasts are operationally generated by the Copernicus Marine IBI Monitoring and Forecasting Centre (IBI-MFC), covering the Iberia-Biscay-Ireland (IBI) area (spanning 26-56°N and -19-5°E). The CMS IBI-WAV-NRT system (IBI_ANALYSISFORECAST_WAV_005_005, IBI-MFC, 2024a) generates both historical best estimates for the preceding two years and hourly instantaneous forecasts extending to a 10-day horizon, with daily updates. IBI-WAV is based on the Météo-France Wave Model (MFWAM), which itself is an adaptation of the ECWAM spectral wave model (present version: IFS-ECWAM-CY47R1, ECMWF, 2020). The model employs a horizontal grid resolution of approximately 1/36° (≈2.5 km). The wave spectrum is discretized into 36 directional bands and 30 frequency bands, covering the range from 0.035 Hz to 0.56 Hz. The model bathymetry is derived from the ETOPO1 global bathymetry dataset, which has a native resolution of 1 arc-minute (1/60°), and is smoothed for numerical stability within the domain.

Wave physics within IBI-WAV are parameterized using the ST4 formulation (Ardhuin et al., 2010), accounting for energy dissipation due to wave breaking and swell decay. The MFWAM implementation of ST4 is further enhanced by incorporating a Phillips tail spectrum to accurately represent the high-frequency portion of the wave spectrum. The $\beta_{max}$

tuning parameter, that adjusts the transfer of energy and momentum from the wind to the waves, is 1.48. This is consistent
with other applications in the literature (i.e. 1.39 in Valiente et al., 2023; 1.52 in Ardhuin et al., 2010 or 1.75 in Alday et al., 2021).

The model is forced with hourly wind fields from the ECMWF-IFS (ECMWF, 2024), provided at a horizontal resolution of 1/8°. Open boundary conditions for wave spectra are obtained from the Copernicus Marine Global Wave Analysis and Forecasting System (product GLOBAL_ANALYSISFORECAST_WAV_001_027, GLO-MFC, 2024), which has a spatial
resolution of 1/10°.

In addition to wind forcing, IBI-WAV incorporates offline surface currents as forcing terms, derived from the IBI Physics Analysis and Forecasting system (product IBI_ANALYSISFORECAST_PHY_005_001, IBI-MFC, 2024b). IBI-WAV includes specific wave-current coupling parameterizations, specifically the surface stress and the generation of turbulence within the oceanic mixed layer due to wave breaking.

Further details on the MFWAM and IBI-WAV can be found in Aouf and Lèfevre, 2015 and Toledano et al., (2022).

Currently, the CMS IBI-WAV NRT service (**Figure 1**) retrieves and processes the following 3 upstream data sources: (FRC-WND) Wind fields from ECMWF-IFS at 1/8°; (FRC-WAV) Boundary Conditions (e.g. wave spectra) from CMS GLO-WAV; (FRC-CUR) Surface currents from IBI-PHY at 1/36°. Once built the forcings, the IBI-WAV model is executed, and the resulting products are disseminated through the Copernicus Marine catalogue.

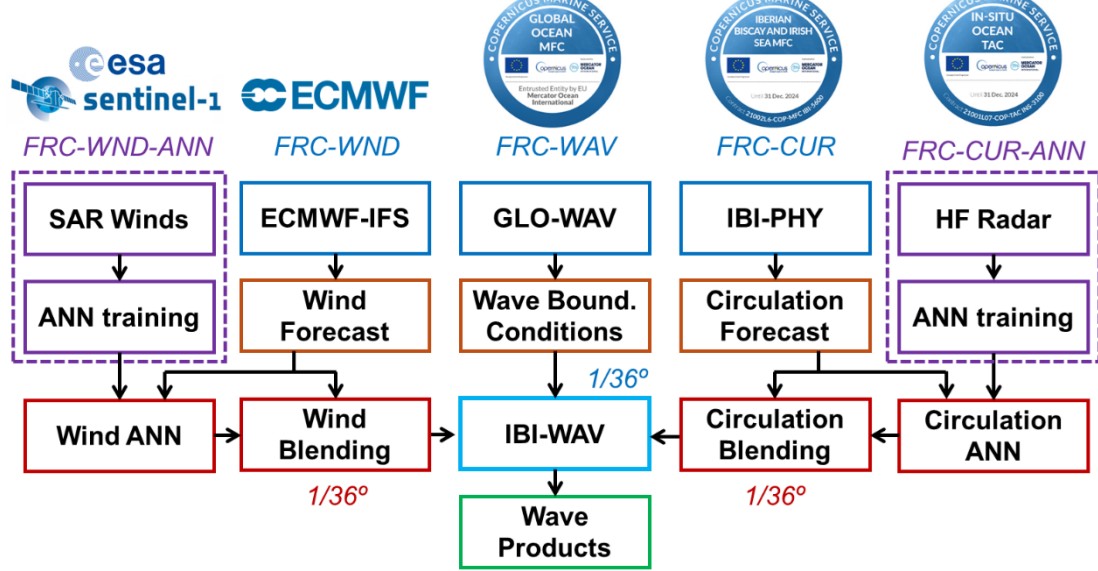


**Figure 1: Flowchart of the proposed upgrades of the KAILANI project for the IBI-WAV operational chain. Note that ANNs training is conducted outside (and offline) the operational chains.**

This article aims to improve the IBI-WAV system by adding two new elements: (i) FRC-WND-ANN and (ii) FRC-CUR-ANN that predicts wind fields/surface currents with Artificial Neural Networks (ANNs) that uses ESA Sentinel 1 SAR/HF

Radar as target dataset. These two new elements are blended with FRC-WND and FRC-CUR; building new forcings at 1/36º resolution.

## 2.2. Generative Adversarial Network for Winds (FRC-WND-ANN)

Super-resolution techniques are used to enhance wind fields by reconstructing fine-scale details from low-resolution data
(Stengel et al., 2020). This methodology is based on three assumptions: (i) the ECMWF-IFS model provides a reliable baseline for wind speed data; (ii) Synthetic Aperture Radar (SAR) data offers comprehensive spatial coverage and is well-suited for coastal remote sensing; and (iii) SAR data can be used to mitigate systematic spatial biases due to its accuracy and high resolution products (1/100º), although time-related biases remains unchanged. Generative Adversarial Networks (GANs) (Goodfellow et al., (2016)) are suitable architectures for training ANNs in super-resolution tasks (Xie et al., 2018;
Tran et al., 2020). Three goals are targeted: (i) correct wind speed values; (ii) enhance the spatial resolution, particularly at the land-sea interface, where the performance of the ECMWF-IFS is limited (Bertotti et al., 2012); and (iii) increase the resolution of wind forcing data to match the IBI-WAV model grid (improving from 1/10º to 1/36º) (Bresson et al., 2018). The GAN architecture consists of two nested networks (i.e. a generator and a discriminator) both of which are built using a chain of Convolutional Neural Networks (see **Fig. 2**). The generator uses ECMWF-IFS (1/10º) winds as a baseline (i.e.
network input) for generating consistent high resolution wind fields (1/100º, network output). Local patterns are added to the original ECMWF-IFS by adding features learnt "a priori" from ESA OCN L2 S1A/S1B wind fields (i.e. target dataset, see below). Besides, the discriminator aims to distinguish a given generator output from the target dataset. As the training progresses, the generator learns how to "mimic" the target dataset, in order that the discriminator is no longer able to distinguish the origin of the data. This procedure makes feasible to train deep ANNs (i.e. several layers chained), and there
are already applications in climate and the ocean (Zhang et al., 2020, Leinonen et al., 2020, Ravuri et al., 2021, François et al., 2021).

At a more detailed level, the generator is composed of two main sections. First, multiple blocks of convolutional layers extract low-resolution features from ECMWF-IFS data. These are then followed by convolution-resize blocks that upscale the data. This upscaling simultaneously reduces the checkerboard effect (Odena et al., 2016) often present in sub-pixel and
transpose convolution layers. The skip layers transfer extra information to layers that were not directly connected, allowing a deeper design that helps with the generalization of the solution. Finally, the generator losses follow the use of Physical Informed Neural Networks (PINNs, see Karniadakis et al., 2021 for a review) aiming to preserve: (i) wind magnitude, (ii) wind divergence and curl, and (iii) directional deviation.

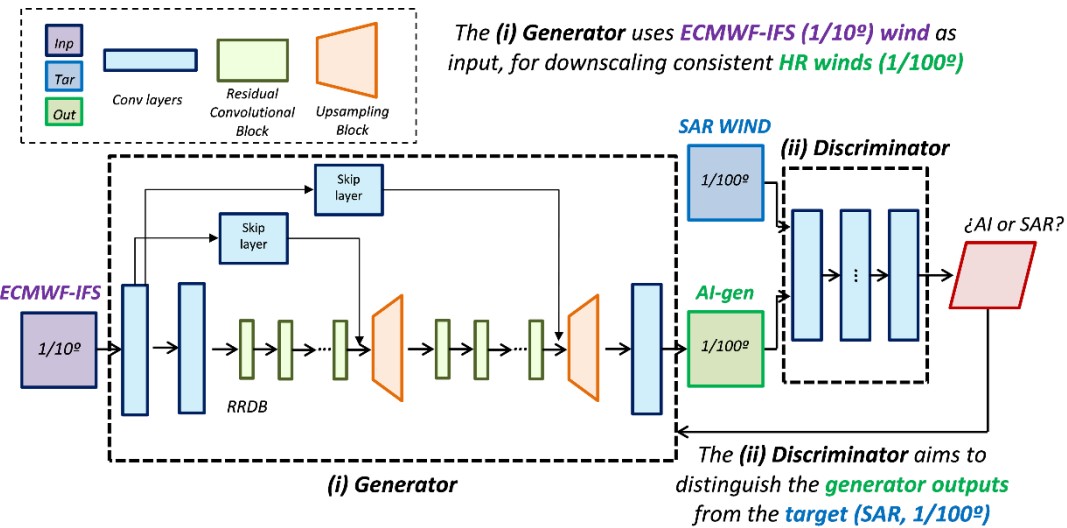

**Figure 2: Generative Adversarial Network (GAN) for Wind Forecast that consists of two parts: (i) a Generator that uses ECMWF-IFS data (purple box) for building 1/100° images (green box), trying to mimic SAR Wind data (blue box), and (ii) a Discriminator (that predict the likelihood that a given image is a true SAR Wind, or rather an AI-generated image).**

The target dataset consists of ESA Ocean Level 2 products (ESA OCN L2 S1A/S1B, 1km resolution) derived from Sentinel 1A/1B SAR images (Mouche et al., 2019, Hajduch et al., 2022). However, due to the satellite revisit time (6 to 12 days, depending on S1A/B availability), there are not enough data at a given region for generalization. Thus, it has been retrieved data from the pilot sites and other places of the continental Europe coastline (mainly NE Atlantic), classifying this data in 3°x3° regions. To prevent degradation of the Artificial Neural Network (ANN) from land effects, images with over 20% land coverage were discarded. In this sense, a relevant human-intensive effort has been devoted in the data curation for the training/testing datasets (i.e. removing spurious pixels, spikes, etc). The training dataset covers from 2018 to 2023, except the testing period (Autumn 2021, same as the Benchmark, see **Subsection 2.4**).

**Blending ANN winds into ECMWF-IFS forecasts**

The Wind ANN generates 1°x1° tiles of wind data at high resolution (1/100°). Integrating this data into the wind forcings used by the IBI-WAV model presents a difficulty: simply cropping the trained regions from the original data and replacing them with the ANN output creates discontinuities in the wind fields. These sharp boundaries are expected to cause instabilities in IBI-WAV and result in spurious wave products.

The blending solution uses 2D window functions with a 2nd order spline interpolation of the cropped tiles (Press, 2007). This method allows to weight grid points when integrating them seamlessly in a larger domain. Hence, it can be blended wind tiles that span the whole IBI area.

## 2.3. Autoencoder for Surface Currents (FRC-CUR-ANN)

Predicting surface currents is more challenging than wind forecasting, particularly in coastal zones. Near the coast, the skill of physical models decreases due to the complex interaction of tidal currents, wind-driven circulation, and wave-induced flows. To overcome this, ANNs built on Autoencoders (AEs) are used to capture the spatio-temporal patterns in HF-Radar (HFR) data (**Figure 3**).

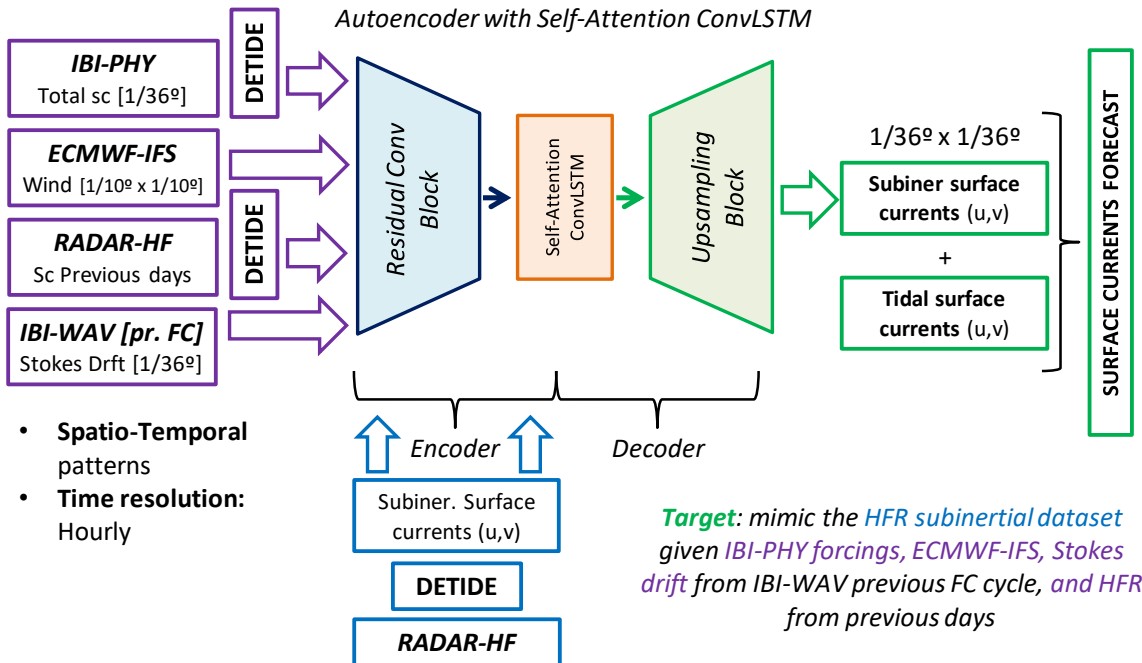

**Figure 3: Architecture for the Surface Currents ANN. Purple boxes denote the input data; blue boxes for target data (HFR) and green ones, resulting products.**

Autoencoders (AEs) are neural networks that learn data representations by mapping input data into a lower-dimensional space and then reconstructing it. This capability allows AEs to extract salient features from HF-Radar (HFR) data and identify their interrelationships. The AE architecture is defined by two key components: (i) an encoder, which compresses the data into a feature subset, and (ii) a decoder, which reconstructs the output dataset to match the characteristics of the original input. The encoder includes Convolutional layers and ConvLSTM layers with self-attention (Shi et al., 2015, Vaswani et al., 2017), that can extract spatio-temporal features with both global and local dependencies. The spatial patterns are learnt with the convolutional layers and the time dimension is addressed with Long-Short Term Memory (LSTM) layers (Hochreiter and Schmidhuber, 1997). LSTM are a specific type of Recurrent Neural Network (RNN) designed to learn long-term dependencies in sequential data by feeding the output of a previous step as input to the current step. This capability is crucial because the very nature of surface currents involves temporal structures and long-term dependencies that are key factors for proper modeling.

The input variables in the AE are (i) subinertial currents forecast from IBI-PHY, (ii) ECMWF-IFS wind forecast, (iii) Stokes Drift forecast from the previous IBI-WAV forecast cycle, and (iv) if available, HFR from previous days (one-week moving window). The target variables are the total zonal and meridional components of the HFR surface currents, retrieved from Copernicus Marine In-Situ TAC (INS-TAC, 2024). Although total currents are inherently smoother than radial ones, they still contain spikes and spurious values, which difficults the Autoencoder (AE) training. However, the strong periodicity of astronomical tides could hinder AE performance. Therefore, both the IBI-PHY and HF-Radar (HFR) data are detided in the preprocessing stage. For consistency, the same tidal constituents included in the IBI-PHY solution were extracted (M2, S2, N2, K2, K1, O1, Q1, P1, M4, MM, and MF; see IBI-MFC, 2024b). Time resolution of both input and target variables is hourly.

**Blending ANN surface currents into IBI-PHY forecasts**

Once the subinertial surface currents were predicted for the three pilot sites, tidal currents were calculated using harmonic analysis (Foreman et al., 2009) with HF-Radar (HFR) tidal constituents and subsequently added. These combined currents were then integrated into the IBI domain using a simplified momentum conservation equation. The blending process involves the following steps: (i) The ANN-generated current forecasts are used directly, without smoothing or modification. (ii) Blending occurs within a buffer zone (a "halo") around the ANN domain. The halo's width varies depending on location and hydrodynamic conditions. Predictions within the halo are calculated using the simplified momentum conservation equation, with boundary conditions provided by the ANN at the inner halo edge and the IBI-PHY model at the outer edge. (iii) Outside the halo, the IBI-PHY solution is unchanged, assuming the ANN's influence becomes negligible at a sufficient distance.

This blending procedure is repeated hourly for the entire forecast period. The computational cost is low, requiring only a few minutes.

## 2.4. Pilot sites and benchmark period

The previous subsections described the IBI-WAV NRT system and the proposed methodology. This section will now focus on the three pilot sites in the IBI area where the methodology was tested: (i) Galicia, (ii) Tarragona, and (iii) Gran Canaria (**Fig. 4**). Following the description of the sites, the benchmark period will be briefly detailed.

The Galician coast, located in northwestern Iberian Peninsula and directly exposed to the North Atlantic, is characterized by a dynamic wind and wave regime. Prevailing westerly and southwesterly winds, driven by the mid-latitude westerlies, are the dominant atmospheric forcing, contributing significantly to regional precipitation (Lorenzo et al., 2008). Seasonal wind variations are pronounced: winter storms frequently reach gale force, while summers experience lighter, more variable winds influenced by the Azores High (Trigo et al., 2002). The complex coastal geomorphology, featuring rias and headlands, generates localized wind phenomena, including coastal jets, land-sea breezes, and katabatic winds. Annual mean wind speeds range from 7 to 9 m/s, with higher values observed in exposed coastal areas and during winter (Herrera et al., 2005). The annual mean significant wave height is estimated at 1.5 to 2.5 meters; during winter storms, wave heights can exceed

several meters (Lorente et al., 2017). Long wave periods, typically 8 to 12 seconds, reflect the influence of distant swell and Atlantic storms. Coastal circulation is primarily driven by tides (macro-tidal environment), wave-induced currents and the northward-flowing Iberian Poleward Current (IPC), which transports warm, saline waters along the continental slope (Péliz et al., 2005). This flow is further modulated by wind forcing, particularly during upwelling events, and by the complex continental shelf topography.

The Tarragona region, situated along the northwestern Mediterranean Sea, experiences a distinct wind and wave regime. Wind patterns result from a complex interaction of large-scale weather systems and local influences. The strong, cold, and dry northwesterly Mistral is a dominant wind, particularly in winter and spring (Guenard et al., 2005). Other significant winds include the Tramuntana (northerly) and Garbi (warm southwesterly), further modulated by local land-sea breezes (Campins et al., 1995). In fact, channelized extreme in-land winds (i.e. wind-jets) are common at the area (Grifoll et al., 2016) and they contribute to wind-sea waves. The wave climate is generally less energetic than Atlantic-exposed coasts but exhibits considerable variability. Mean significant wave height typically ranges from 0.5 to 1.5 meters, increasing up to five-fold during storm events (Bolaños et al., 2009). Mean wave periods generally range from 4 to 8 seconds, reflecting the fetch-limited and less intense storms compared to the Atlantic. Wave direction is predominantly east and southeast, varying with prevailing wind conditions. Coastal circulation is primarily influenced by the general cyclonic circulation of the Western Mediterranean as the tidal range is low (microtidal), with a tendency for southward alongshore currents (Millot, 1999). This circulation is locally modified by wind forcing, waves, the Ebro River discharge, and the complex coastal geometry (Lorente et al., 2021).

Gran Canaria, in the Atlantic Ocean off the northwest African coast, is influenced by its subtropical latitude, proximity to Africa, and volcanic topography. Prevailing winds are dominated by the relatively consistent northeast trade winds, especially during summer. These trade winds, driven by the Azores High, are a key feature of the subtropical North Atlantic. Winter sees increased influence from mid-latitude weather systems, leading to more variable wind directions and occasional storms (Fernandopullé, 1976). The island's mountainous terrain generates significant local wind variations, including acceleration/deceleration zones, lee effects, and katabatic winds on the leeward side. The wave climate combines locally generated wind waves and North Atlantic swells. Mean significant wave height ranges from 1 to 2 meters, reaching nearly 5 meters during winter storms and swells. Wave periods typically range from 6 to 10 seconds, reflecting both local winds and distant storm influence (Semedo, 2018). Wave direction is predominantly north and northeast, consistent with the trade winds and swell. Coastal circulation is influenced by tides (meso-tidal) and the southward-flowing Canary Current (Barton et al., 2001), a branch of the North Atlantic Current, and is further modulated by local wind forcing, waves and island-induced eddies (Arístegui et al., 1994).

**Figure 4** shows the main features at each site (e.g. available observational network and the bathymetry); whilst **Table 1** summarizes the coordinates and typologies for each water sensor. There are buoys as Villano (GAL-D) that will be analysed for waves, winds and currents; but others will be addressed only at specific sections.

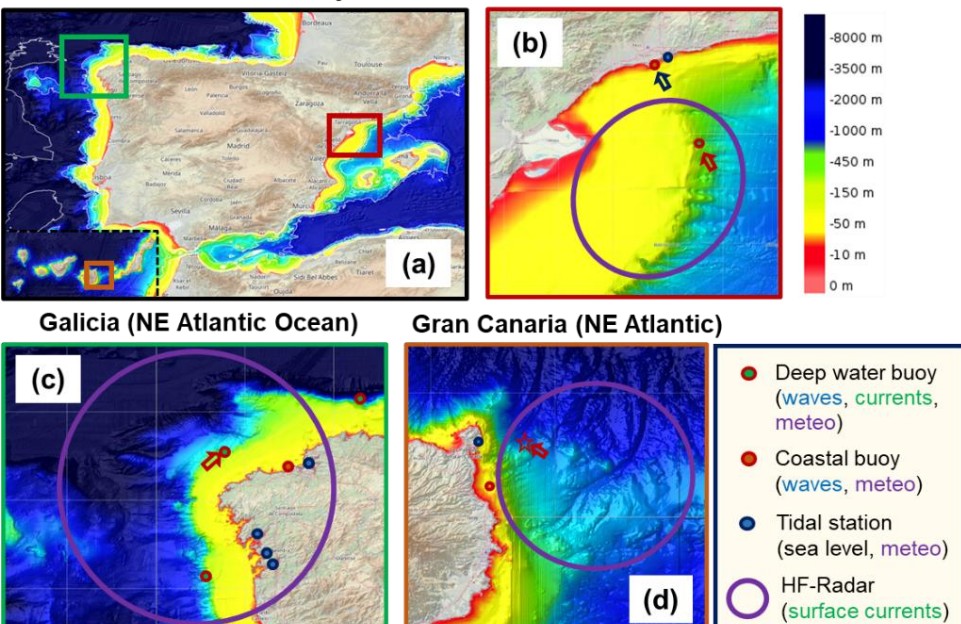

**Figure 4: Pilot sites at the IBI area [a]: Tarragona [TAR] [b], Galicia [GAL] [c] and Gran Canaria [GCA] [d]. The coloured areas are the bathymetry from EMODNET, and the colorbar shows the water depth (in m). The icons show the available observational network. Red arrows show the position of the wind/subinertial currents analysis points. The deep blue arrow in [b] is the position of the coastal buoy.**

| Name | Description | Water depth [m] | Longitude | Latitude | Wave | Wind | Currents |
|------|-------------|-----------------|-----------|----------|------|------|----------|
| **GCA-C** | *Gran Canaria Coastal Buoy* | 30 | *15.39ºW* | *28.05ºN* | X | | |
| **GCA-D** | *Gran Canaria Deep Buoy* | 780 | *15.80ºW* | *28.20ºN* | X | | |
| **GCA-P** | *Gran Canaria HFR point* | 1400 | *15.33ºW* | *28.17ºN* | | *X* | *X* |
| **TAR-C** | *Tarragona Coast* | 15 | *1.19ºE* | *41.07ºN* | X | | |
| **TAR-D** | *Tarragona Deep* | 688 | *2.20ºE* | *41.32ºN* | X | X | X |
| **GAL-C** | *Gijón Coastal Buoy* | 54 | *5.66ºW* | *43.62ºN* | X | | |
| **GAL-D** | *Galicia - Villano Deep Buoy* | 386 | *9.21ºW* | *43.50ºN* | *X* | *X* | *X* |

**Table 1: Coordinates and water depth of the points that will be analysed in this section. The ID font colour refers to each pilot site (dark orange, Gran Canaria; crimson, Tarragona; green, Galicia). The columns wave, wind and currents show which EOVs are measured by each buoy.**

At Tarragona and Galicia there are deep-water buoys within the HFR coverage (see red arrows in **Figure 4** [b,c]). This allowed us to verify the system consistency with in-situ data. However, at Gran Canaria, the deep-water buoy (**Table 1**) is outside the range of the HFR (NW part of the island, whilst the HFR is in the Eastern part). Results will be shown at a point

in a specific area that we detected that IBI-PHY does not perform properly (red star in **Figure 4**[d]), and we have availability of HF-radar data. The area near Gran Canaria harbor poses a challenge because its bathymetric gradients are too steep to be properly modeled with the current IBI-PHY resolution (1/36º). As a result, probable pressure-gradient inconsistencies (García-León et al., 2022) lead directly to a persistent overestimation of surface currents.

The benchmark period was selected from the full analysis period (January 2021 to January 2023) based on criteria designed to isolate model underperformance. Specifically, a period was chosen if the Coefficient of Efficiency (COE) (Legates and McCabe, 2013) for the IBI-WAV wave parameters was below both a specified threshold and the average performance. Consequently, the benchmark period ranges from September 2021 to January 2022. This period showed low performance across all three pilot sites.In the NW Mediterranean, this period was featured by short-duration moderate events mixed with calm periods. At Tarragona, a relevant share of the wind regimes was in-land (W-NW), mainly influenced by topographic local constraints that ECMWF-IFS coarse resolution cannot properly represent (Cavalieri et al. 2024). The period featured unusual storms at the North-East Atlantic, such as (i) the storm Arwen (25[th] – 27[th] November 2021) and (ii) the January 2022 storm (20[th] - 23[rd] January 2022), which are briefly described below:

(i) Storm Arwen, a powerful extratropical cyclone, impacted the UK and parts of continental Europe around 26[th] November bringing strong winds, heavy snow, and blizzards, especially across Scotland and northern England. In Galicia, the storm produced 6.5m waves and 14 m/s winds. Arwen's rapid intensification stemmed from strong temperature gradients between Arctic and warmer air, creating baroclinic instability. This, combined with an upper-level trough, fuelled rapid intensification, marked by a plunging central pressure and surging wind speeds. A blocking high at the west of Ireland forced the storm southwards instead of the typical eastward track, tightening the pressure gradient on its eastern side and generating strong northerly winds.

(ii) The January 2022 episode, J-22 hereafter, had a synoptic pattern featured by a high-pressure system positioned over the British Isles and a low-pressure system located south of the Azores, generating moderate wind-sea waves near the three pilot sites. In Galicia, easterly winds across the Cantabrian Sea created significant waves, peaking near 4 meters with 6-second periods. Wave heights exceeded 3 meters for about 24 hours on 22[nd] January, alongside sustained above 10 m/s winds, with a maximum recorded height of 7 meters. Tarragona experienced northerly Tramontana winds leading to NE waves around 1 meter. Near Gran Canaria, moderate southwest winds generated south-eastward waves of about 1.8 meters with a 5-second period. While the wind generation area was south of the islands, they shielded the eastern coast (where the buoy was located) from the strongest southwest winds, keeping wind speeds below 5 m/s.

Both storm events will be analysed in the next Sections, for quantifying the preliminary impact of the methodology under extreme regimes.

## 2.5. Testing of the benchmark period

To evaluate the KAILANI methodology's feasibility, a series of sensitivity tests were performed using the IBI-WAV system
with different combinations of forcings. The tests span the whole benchmark period (i.e. Autumn 2021, see Sec. 2.4). All the
experiments use a model set-up analogous to the one in operations, but without data assimilation.

Four experiments were conducted (**Table 2**). The control simulation (CNT) spans from October 2021 to January 2022. The
wave boundary conditions (spectra) are from the operational GLO-WAV NRT. The ECMWF winds and IBI-PHY are also
from the operational set-up.

The WND/CUR experiments are the same than the control run except for replacing the operational winds/surface currents
with ANN-generated counterparts. The TOT simulation incorporated both ANN-generated wind and surface current
forcings. The subsequent analysis considered two aspects: (i) overall performance across the entire benchmark period and
(ii) performance during specific extreme events, namely Storm Arwen and the January 2022 storm.

| Exp | Description |
|-----|-------------|
| **CNT** | *Control (winds: ECMWF-IFS [1/8°], currents: IBI-PHY [1/36°])* |
| **CUR** | *Winds: ECMWF-IFS [1/8°], currents: ANN [1/36°]* |
| **WND** | *Winds: ANN [1/36°], currents: IBI-PHY [1/36°]* |
| **TOT** | *Winds: ANN [1/36°], currents: ANN [1/36°]* |

**Table 2: List of experiments for assessing the performance of KAILANI. Exp: Acronym of the experiment. Description: list of
forcings used at each experiment. Benchmark period: 2021-10/2022-01.**

## 2.6. Error metrics

The performance of the ANNs and the IBI-WAV tests have been assessed with a set of error metrics that are briefly
summarized in this Subsection. In all cases, $P_i$ and $O_i$ refer to the computed and observed signals, respectively; N is the
number of time points and $(\bar{\cdot})$ is the mean operator.

The Bias (Eq. 1) represents the integrated error between predicted and observed signal.

$$Bias = \frac{1}{N}\sum_{i=1}^{N}(P_i - O_i) , \qquad (1)$$

The Root Mean Square Deviation (RMSD, Eq. 2) represents the sample standard deviation between predicted and observed
signal.

$$RMSD = \sqrt{\frac{1}{N}\sum_{i=1}^{N}(P_i - O_i)^2} , \qquad (2)$$

The Correlation coefficient (Corr, Eq. 3) is a measure of the linear correlation between two signals. It ranges between +1 and
-1, where 1 is total positive linear correlation, 0 is no linear correlation, and -1 is total negative linear correlation.

$$Corr = \frac{\sum_{i=1}^{N}\left((P_i-\bar{P})(O_i-\bar{O})\right)}{\sqrt{\sum_{i=1}^{N}(P_i-\bar{P})^2}\sqrt{\sum_{i=1}^{N}(O_i-\bar{O})^2}}, \tag{3}$$


The Coefficient of Efficiency (COE, Eq. 4) (Legates and McCabe, 2013) is a measure of model performance, with a value of 1 representing a perfect fit. A COE of 0 indicates that the predictive capacity of the model equals to using the mean of the observed values. Therefore, a negative COE signifies the model performs worse than the measured mean.

$$COE = 1 - \frac{\sum_{i=1}^{N}|O_i - P_i|}{\sum_{i=1}^{N}|O_i - \bar{O}|}, \tag{4}$$

The Scatter Index (SI, Eq. 5) indicates how wide the difference between the modelled data and the observed data is scattered relative to the mean of the observations. The units are in percentage.

$$SI = \sqrt{\frac{\sum_{i=1}^{N}\left((P_i-\bar{P})(O_i-\bar{O})\right)^2}{\sum_{i=1}^{N}O_i^2}}, \tag{5}$$

## 3 Results

This section presents the assessment of the methodology of KAILANI over the benchmark period (see **Subsection 2.4**). Specifically, **Subsection 3.1** details the results obtained from the Wind ANN (introduced in **Subsection 2.2**). **Subsection 3.2** then presents the corresponding results for the Surface Current ANN (described in **Subsection 2.3**). Finally, **Subsection 3.3** summarizes the sensitivity tests conducted on the IBI-WAV system (**Subsections 2.1 and 2.5**).

### 3.1. Wind ANN generated forcings

Three points needs to be addressed to evaluate the Wind ANN performance: (i) verify that the SAR data has better skill than the ECMWF-IFS, (ii) demonstrate that the ANN-generated winds achieve comparable accuracy to the SAR data, (iii) quantify that the ANN error metrics are lower than IFS.

The analysis confirmed that SAR data generally exhibits superior skill compared to the ECMWF-IFS wind speed product. At Galicia, the ECMWF-IFS bias was notably higher (1.35 m/s) than the SAR bias (0.96 m/s). Furthermore, SAR demonstrated

greater accuracy in terms of RMSD (1.49 m/s vs. 1.93 m/s) and Scatter Index (SI) (56.1% vs. 59.2%). Tarragona followed a similar pattern: ECMWF-IFS continued to overestimate wind speed (1.05 m/s vs. 0.71 m/s in SAR) and had a higher RMSD (1.92 m/s vs. 1.24 m/s in SAR). The only exception was the SI, which was marginally higher for SAR (74.0%) than for ECMWF-IFS (73.1%).KAILANI shows better metrics than ECMWF-IFS, remarkably in RMSD. As an example, **Figure 5** shows the performance of the ECMWF-IFS and the Wind ANN at Galicia. Despite that the IFS skill is good (**Figure 5[b]**,

blue rectangle), the ANN present better metrics (**Fig. 5[a]**, red rectangle): (i) the data is more clustered along the 1:1 line, with reductions of the scatter-index close to 3%, (ii) the extremes (speeds above 15 m/s) are better reproduced (IFS tends to

underestimate them). The Taylor diagram (**Fig. 5[c]**) confirms the same trend, with increases of the correlation (IFS: 0.97 and ANN: 0.99) and lower RMSD (IFS: 0.57m/s and ANN: 0.38 m/s). So, it can be concluded that the ANN is able to predict winds that are closer to the SAR data than IFS.

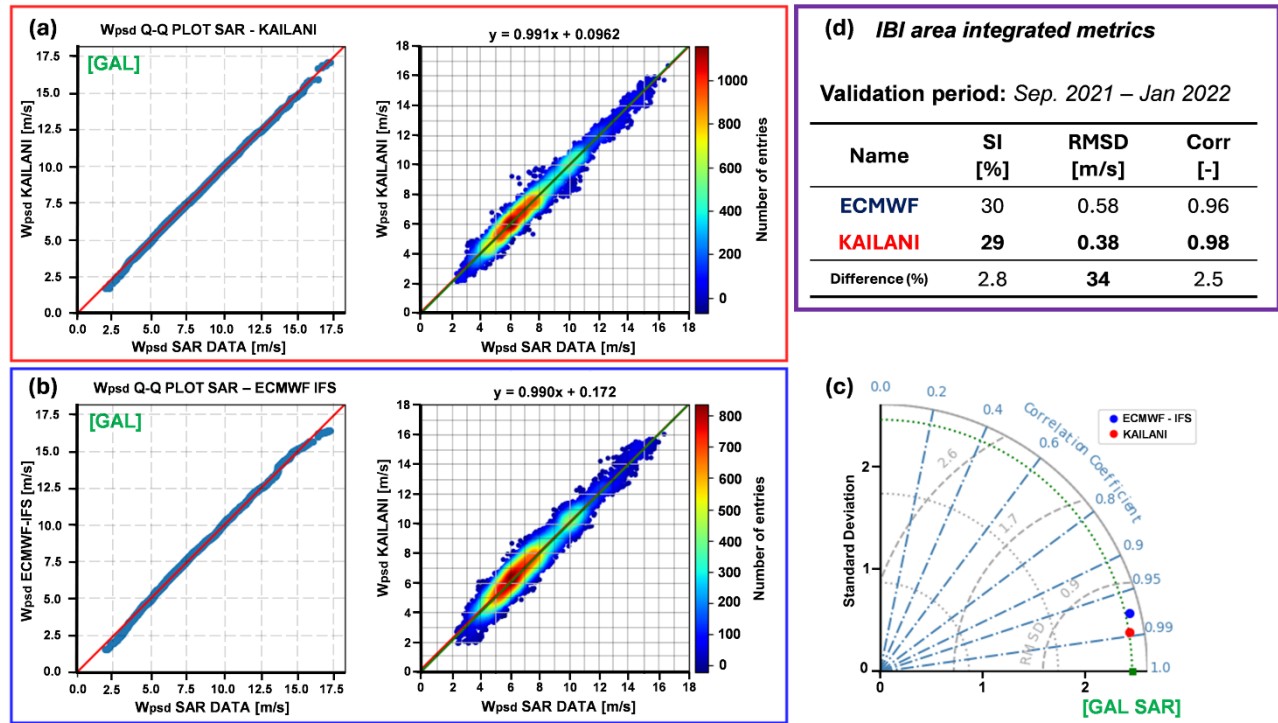

Figure 5: [a-c] Wind ANN results for Galicia (GAL) Pilot Site. OCN L2 SAR Winds at the Galicia area vs [a] KAILANI Wind-ANN solution (red); and [b] original ECMWF-IFS data (blue). Common to [a-b] subplots: (Left) Q-Q plot of SAR inferred wind speed (x-axis) vs. [a] KAILANI / [b] ECMWF-IFS (y-axis) at site. (Right) Scatter-plot of SAR inferred wind speed (x-axis) vs. [a] KAILANI / [b] ECMWF-IFS (y-axis). The green solid line shows 1:1 ratio; whereas the red one the linear fit to both datasets. Title of the subplot denotes linear fit equation. The colours show density of entries for a specific pair value. [c] Taylor diagram of the same KAILANI and ECMWF-IFS, respect to the SAR data. [d] Integrated error metrics at the IBI area.

The results shown in Galicia are consistent across the different analysed zones of the IBI area. Note that the metrics (and their differences) shown in the Taylor diagram (**Fig. 5[c]**) are similar than the spatial-average across the region (**Fig. 5[d]**): (i) significant reductions of the RMSD (KAILANI is 34% lower than IFS) and moderate improvements on Scatter-index (2.8%) and correlation (2.5%).

The spatial variability of the wind speed RMSD is shown in **Fig. 6**. Each colored dot represents the centroid of a 3º×3º region, and each region provided approximately 3000 SAR data samples used for training the Wind ANN. In general, the GAN tends to perform better (achieving lower RMSD) in the coastal zone than in offshore waters, because the training dataset contained more samples close to the coast. RMSD reductions (IFS vs KAILANI) close to 35% are found along the Atlantic French coastline, the Cantabrian Sea and Tarragona. However, this reduction reaches around 25% in the Portuguese coast and the Gibraltar strait.

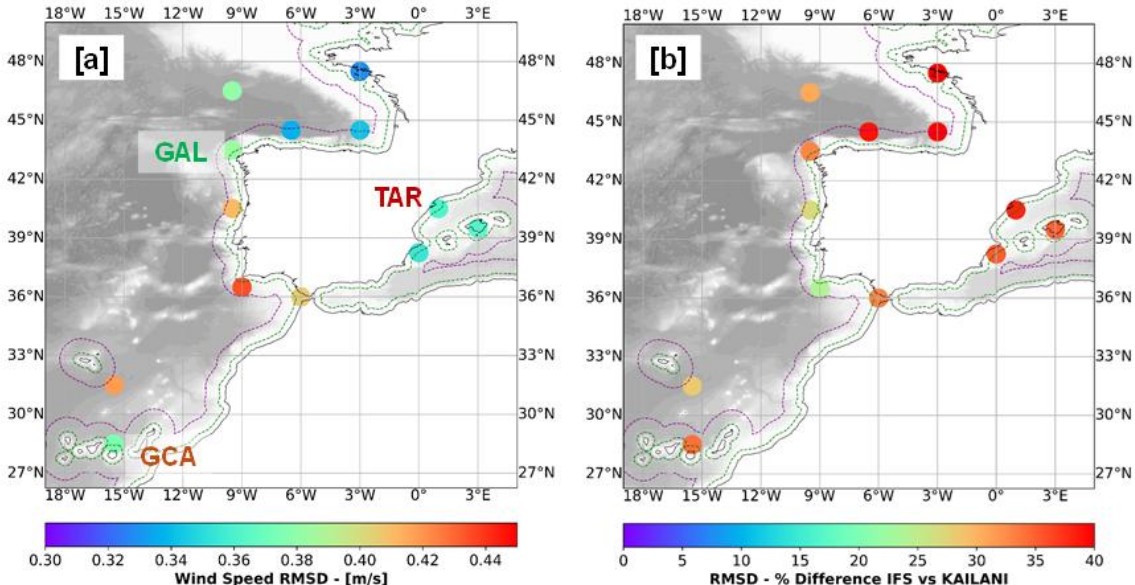

**Figure 6: Distribution of the RMSD at different zones of the IBI area. The coloured dots denote [a] RMSD obtained with KAILANI. [b] Increment of performance (in %) from KAILANI respect the ECMWF-IFS winds. Dashed green/purple lines display the 30 km/1° zones from coastline.**

As outlined in **Section 2.2**, ANN winds are generated in 1°x1° tiles at 1/100° resolution across the entire IBI-WAV domain. These tiles must be blended to ensure spatial continuity in the final wind field. **Figure 7** illustrates the blending process using an example wind field. While the unblended approach (**Fig. 7[a]**) introduces unrealistic discontinuities at the tile boundaries, the blended result (**Fig. 7[b]**) is smooth and successfully reconstructs realistic wind patterns. The values at each tile are the same in both images, with only differences in the boundaries of each tile. While the unblended approach looks unrealistic and not valid for wave modelling, the blended fields reconstruct consistent wind patterns.

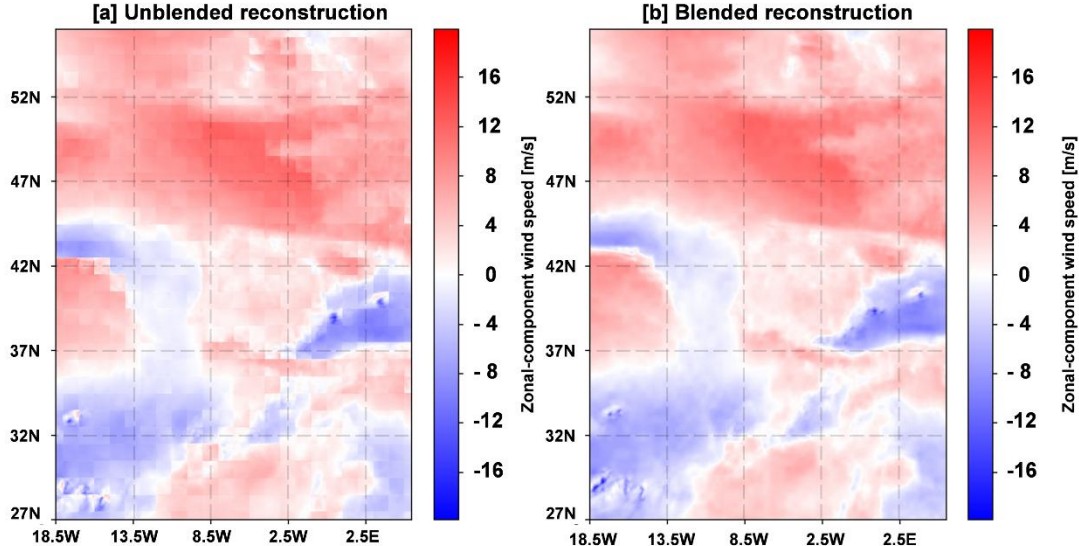

**Figure 7: Comparison between [a] Unblended and [b] Blended reconstruction of E-W component of wind speed by using 1°x1° tiles in the IBI region. At both cases, the tiles have been generated with the same wind ANN. The x-y axes show the indexes of the pixel. As a reference, The S-W corner shows the Canary Islands [18.5ºW,26ºN], the N-E corner [5ºE ,56ºN]. Spatial resolution: 1/36º.**

## 3.2. Surface currents ANN

This subsection will present the results for the ANN Surface Currents. Three separate ANNs were trained, one for each pilot zone. That differs from the one only model for the Wind ANN, that included training data both from the North-East Atlantic and the NW Mediterranean. Each ANN has been tested at representative points for each pilot site (see **Table 1**). Error metrics for the subinertial current speed and direction ($C_{spd}$ and $C_{dir}$) can be found in **Table 3** and **4**, respectively. **Figure 8** shows the current roses for the two models and the HFR; and **Figure 10** denotes the QQ and scatter plots for the $C_{spd}$ at TAR and GAL.

| Name | N | Bias | | RMSD | | Correlation | | SI | |
|------|---|------|------|------|------|------|------|------|------|
| | | $C_{spd\_IBI}$ [cm/s] | $C_{spd\_KAI}$ [cm/s] | $C_{spd\_IBI}$ [cm/s] | $C_{spd\_KAI}$ [cm/s] | $C_{spd\_IBI}$ [-] | $C_{spd\_KAI}$ [-] | $C_{spd\_IBI}$ [%] | $C_{spd\_KAI}$ [%] |
| GAL | 3600 | 3.9 | **-0.8** | 11.8 | **8.4** | 0.32 | **0.35** | 90.6 | **54.6** |
| GCA | 3528 | 5.5 | **1.2** | 12.8 | **9.9** | 0.34 | **0.37** | 106.6 | **67.2** |
| TAR | 3048 | -3.9 | **-2.8** | 16.9 | **14.3** | 0.19 | **0.31** | 56.3 | **50.4** |

**Table 3 Spatially averaged error metrics (HF-Radar vs. IBI-PHY vs. ANN) at the HF-Radar areas of the pilot sites. Covered period: 2021-09 / 2022-01. Columns: name of the buoy; (N) number of time-points for the error metrics, bias for the subinertial current speed ($C_{spd}$) for the IBI-PHY (IBI) and KAILANI (KAI); same for the Root Mean Square Deviation (RMSD), correlation and Scatter Index (SI). Bold numbers denote the best performance.**

The AE improved all the metrics at the three sites, with similar accuracy: (i) speed and directional bias are close to 2 cm/s
and 6º, respectively; (ii) speed RMSD is close to 11 cm/s and (iii) correlation close to 35% and 50%. The AE performance
under regime shifting episodes tends to improve with the previous day observation as input. Also, adding the Stokes drift
increases the skill under extreme regimes (e.g. storm-waves).

The current roses (**Figure 8**) and the QQ-plots (**Figure 9**) reinforce these metrics. At Galicia, IBI-PHY overestimates HFR
(especially once surpassed 20 cm/s) and the SI is high (91%). At the IBI-PHY rose, there are relevant weight on NNW and N
sector (close to 10%), that are lower in the HFR rose (close to 5%). The AE improves the $C_{spd}$ and $C_{dir}$ biases (reaching -0.8
cm/s and -7º), and the $C_{dir}$ correlation (from 0.23 to 0.62). Despite the $C_{dir}$ bias decreases, there are still certain mismatches in
the directional sectors. For instance, the AE has relevant weight in the NW (close to 15%), WNW (10%) and W (8%).
However, the HFR data has NW (7%), WNW (12%), W (10%). Another issue is that the $C_{spd}$ SI is still high (55%), though
the Q-Q plot shows better fit, especially under extreme currents (i.e. above 40 cm/s, see **Fig. 9[i]**).
Tarragona presents similar performance improvements, but the bias keeps higher than in Galicia (-2.8 cm/s). IBI-PHY $C_{spd}$
correlation was the lowest in the three sites (0.19 vs 0.31) due to inconsistencies in the NW Mediterranean barotropic
transport (Sotillo et al., 2021). The directional patterns, though, are better captured in KAILANI (**Fig. 8[a]**). IBI-PHY
overestimates the frequency at the NNE, NE and ENE sectors. As a trade-off, it underestimates the frequency of the SSE, S
and SSW sectors. The relative frequencies between calm (less than 20 cm/s) and moderate (between 20 and 40 cm/s) are also
better handled. As a drawback, the AE also tends to smooth values under extreme regimes (up to 40 cm/s).

| Name | N | Bias | | Correlation | | COE | |
|---|---|---|---|---|---|---|---|
| | | $C_{dir\_IBI}$ | $C_{dir\_KAI}$ | $C_{dir\_IBI}$ | $C_{dir\_KAI}$ | $C_{dir\_IBI}$ | $C_{dir\_KAI}$ |
| | | [º] | [º] | [-] | [-] | [-] | [-] |
| GAL | 3600 | -46 | **-7** | 0.23 | **0.62** | 0.13 | **0.37** |
| GCA | 3528 | 10 | **7** | 0.22 | **0.45** | 0.07 | **0.26** |
| TAR | 3048 | 21 | **-4** | 0.27 | **0.33** | -0.27 | **0.04** |

**Table 4 Spatially averaged error metrics (HF-Radar vs. IBI-PHY vs. ANN) at the HF-Radar areas of the pilot sites. Covered period: 2021-09 / 2022-01. Columns: name of the buoy; (N) number of time-points for the error metrics, bias for the subinertial current direction ($C_{dir}$) for the IBI-PHY (IBI) and KAILANI (KAI); same for the correlation and Coefficient of Efficiency (COE). Bold numbers denote the best performance.**


Despite the limited sample size for extreme current regimes (above 40 cm/s), the AE presents lower dispersion than the IBI
model (**Fig. 9[ii]**). The AE also exhibits better fitness than IBI across moderate current regimes, and its predictions are
generally more aligned with the 1:1 line. However, the improvement in the SI is moderate (from 56% to 50%). A possible
reason is that the observational sample at Tarragona was 15% lower (3048 vs. 3600 time-points) than the sample at the other
sites. The AE tends to learn that a specific directional sector is dominant (SSW) at Gran Canaria (**Fig. 8**), to the detriment of
other sectors in which the HFR has frequencies close to 10% (SSE, WSW). But it captures the regimes better than IBI (be it
calm, moderate or extreme). IBI tends to overestimate the currents, and the calm currents have lower frequency than the

HFR. This behavior can be explained because the HFR has limited coverage and it is closer to the coast (see **Fig. 4**). IBI-PHY cannot solve the alongshore properly, neither does the associated dissipation due to two reasons: (i) the inner shelf has a steep slope, and (ii) the model resolution is not enough (2.5 km). The AE improves relevantly the Bias in $C_{spd}$ (from 5.5 to 1.2 cm/s), the $C_{dir}$ correlation (from 0.22 to 0.45) and COE (from 0.07 to 0.26).

Note also that the AEs directional distribution during extreme conditions (up to 40 cm/s) are better aligned with the measurements at all 3-pilot sites (**Fig. 8**). This correction of the direction may have relevant impact with moderate and extreme currents, that may coincide with storm waves (e.g. the joint action of storm-surges and extreme waves, Pérez-Gómez et al., 2021).

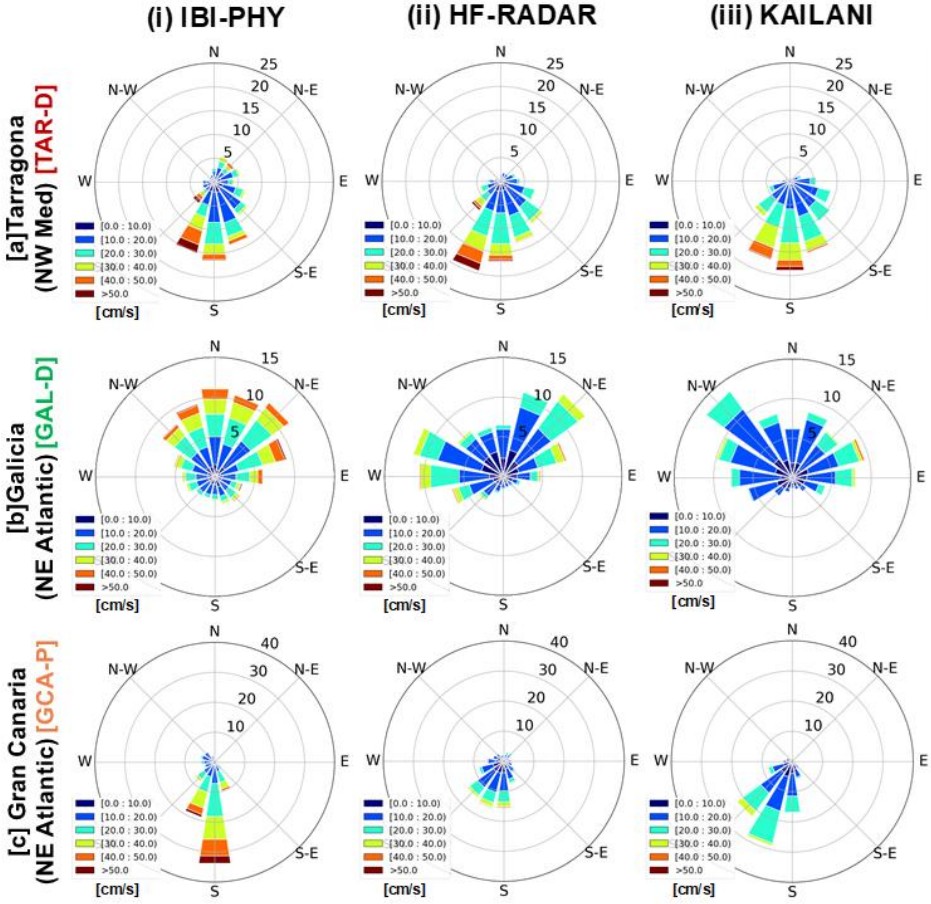

**Figure 8:** Rows: Subinertial current roses at the three pilot sites: [a] TAR-D, [b] GAL-D, [c] GCA-P (Table 1). Columns: 1st (IBI-PHY); 2nd (HF-Radar) and 3rd (KAILANI). The polar plot radius shows the frequency % at a given directional sector. Sectors are binned by 10 cm/s increments (colour value at the legend). Data from the testing dataset. For the sake of comparison, the two models and HF-RADAR have the same number of time points (see Tables 4 and 5). Directional convention: the direction where the currents go (e.g. GCA-P main directional sector will be Southward).

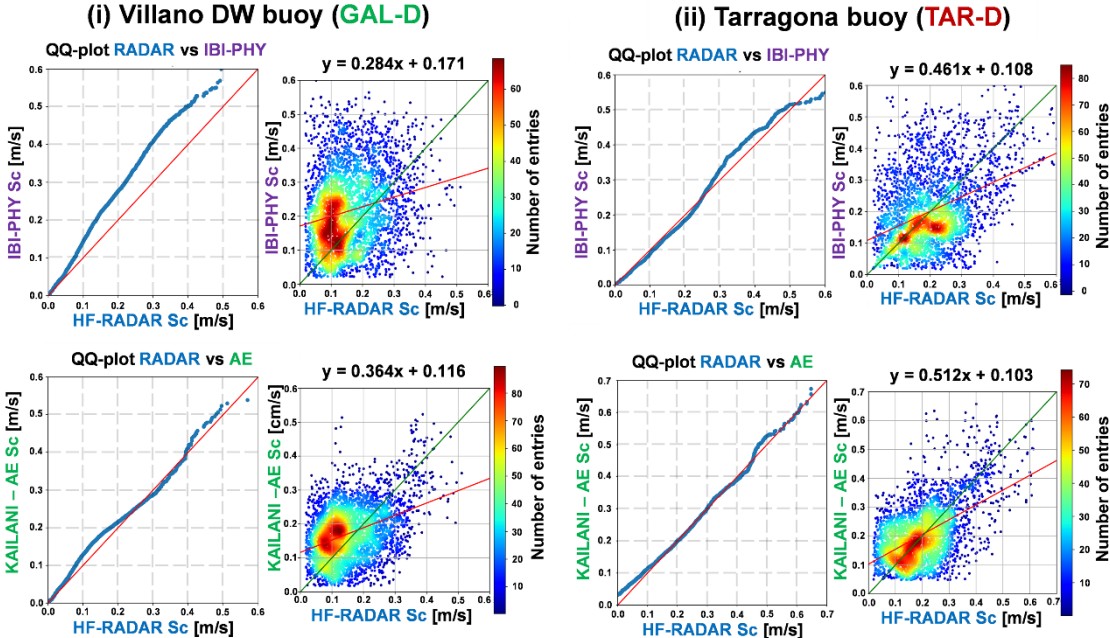

**Figure 9: Rows: Subinertial surface current speed (C$_{spd}$) QQ-plots and scatter-plots from IBI-PHY [a] and KAILANI [b] at Villano (GAL-D) and Tarragona (TAR-D) buoys (see Fig. 2 and Table 1). The format of the figure is the same as in Fig. 5.**


### 3.3. Testing of the benchmark period with IBI-WAV

The previous subsections established the improved performance of the ANN forcings relative to operational forcings during the benchmark period. This subsection quantifies the practical impact of these improvements on the IBI-WAV system. The evaluation will first assess the overall system performance throughout the entire benchmark period, followed by a detailed

analysis of the system's response to the two extreme events identified in **Subsection 2.4**: (i) Storm Arwen and (ii) the J-22 storm.

The general performance of the IBI-WAV system reveals several main patterns. First, wind correction has a greater influence on the solution compared to surface current corrections. This is likely due to the application of wind corrections across the entire IBI domain, whereas current corrections are localized to the pilot sites. While the CUR experiment demonstrates local

improvements at these pilot sites, particularly during specific events, the primary impact of enhanced surface currents is observed in the Mean Wave Period, attributed to Doppler shift effects.

When considering metrics across the entire benchmark period, the CUR run exhibits the closest agreement with the control run (CNT). The WND and TOT runs share strong similarities, as will be further illustrated through the analysis of specific extreme events. Notably, the TOT simulation, incorporating both ANN forcing fields, yields the best overall performance

metrics. **Figs. 10 and 11** showcases the performance of the TOT experiment compared to the control (CNT), highlighting the Cabo de Peñas buoy (Cantabrian Sea, near GAL-D) as the station exhibiting the most significant improvement.

Analysing the impact on Wave Height ($H_{m0}$), the TOT simulation improves both bias and RMSD in the NE Atlantic. However, it leads to degradation in the NW Mediterranean. $H_{m0}$ bias improvements of approximately 10% are observed at GAL (**Fig. 10[a,b]**). At Gran Canaria, the GCA-C buoy experiences a bias degradation of nearly 10%, while the GCA-D buoy shows a corresponding improvement. The bias increases by approximately 20% in Tarragona. The RMSD pattern mirrors the bias pattern, albeit with smaller variations (**Fig. 10 [c,d]**), showing moderate gains at GAL (5%) and losses at TAR (close to 15%).

A similar trend is observed for Mean Wave Period ($T_{m02}$), where the TOT simulation improves bias and RMSD in the NE Atlantic but degrades them in the NW Mediterranean. The $T_{m02}$ bias (**Fig. 11[a,b]**) displays a sharp version of this pattern, with improvements of approximately 25% at GAL and 10% at GCA. However, at TAR, the average degradation reaches 30%. It is worth noting that the RMSD at the Villano buoy (GAL-D) was already the lowest (0.5 s), contrasting with the higher RMSD at GCA-C (1.4 s) (**Fig. 11 [c,d]**). RMSD values in the NW Mediterranean were around 0.7-0.8 s but also exhibited degradation compared to the CNT run.

Overall improvement is also suggested during the two extreme events. However, because the sample of storms is limited, the results and interpretations must be considered as preliminary. **Figure 12** illustrates the expected order of magnitude of the differences between the ECMWF-IFS and the Wind ANN products. These daily-averaged differences are representative of how the Wind ANN corrected the wind forcings during the peak of Storm Arwen. Furthermore, they provide qualitative insight into the reasons for the over- or under-performance observed in the wave results for both the TOT and WND experiments. During storm Arwen, strong Northern winds at the Northern Iberian Peninsula (**Fig. 12[a]**) generated Northern wind-sea waves, that were reinforced with NW swell waves. The integrated $H_{m0}$ of this mixed sea state can be seen in **Fig. 13**. The storm peak was better handled (**Fig. 13[a]**), because at the wind-sea generation area, the ANN Winds predicted more intensity than ECMWF-IFS. The wind speed time series at GAL-D does not show much difference (**Fig. 13[c]**), but **Fig. 12[b]** shows clearly the extra wind energy at the area (red marked region), that lead the TOT and WND simulations to capture better the storm peak at 27[th] November 2021. Note also that the NW Mediterranean has less energy in KAILANI than in IFS (i.e. the blue area in **Fig. 12[b]**), suggesting the same issues mentioned in the general performance.

The CUR simulation was the one that had better skill in the Mean Wave Period during the 24[th] - 25[rd] November (purple shaded slices in **Fig. 13[a,b]**). Surface currents were overpredicted by IBI-PHY, whilst the ANN showed lower bias and variability (**Fig. 13[d]**, and **Subsection 2.4**). Predictions vary significantly across the experiments. Note that in these two specific days, the TOT and CUR runs are closer. Mean wave period decreases with respect to CNT, implying a shifting of wave energy towards higher frequencies, mainly due to Doppler shift.

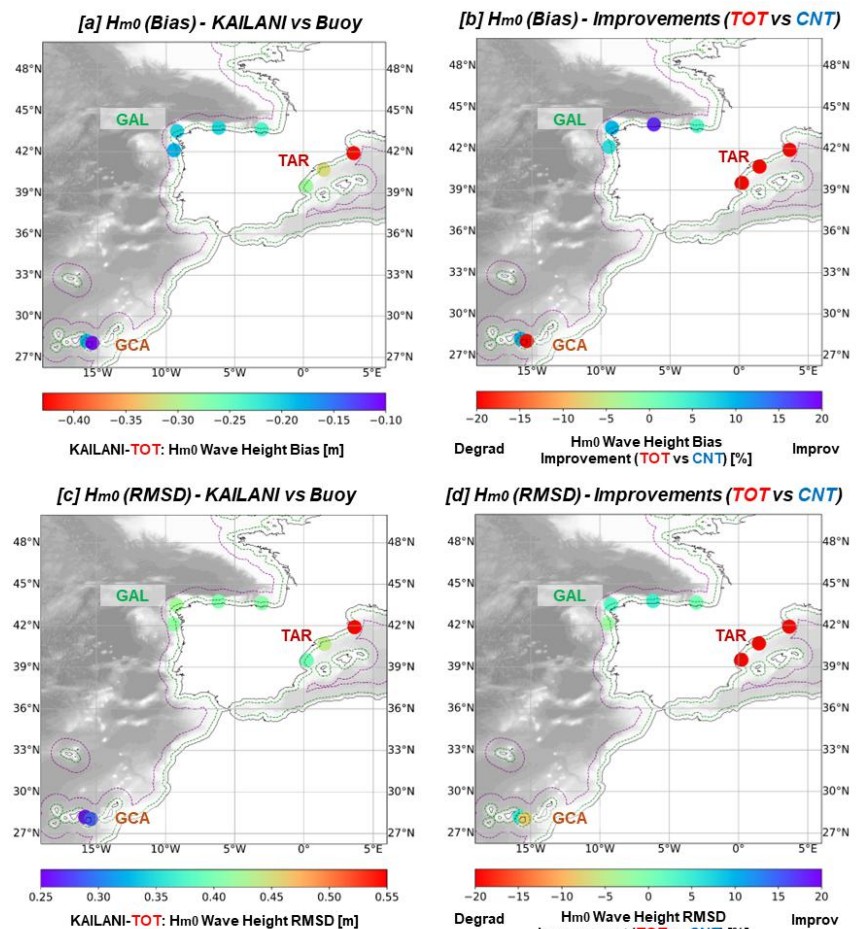

**Figure 10: Integrated $H_{m0}$ error metrics for the whole simulation period (2021/10 - 2022/01). Left panels [a,c] show the Bias and RMSD at the TOT experiment. Right panels [b,d] the % of improvement of the same metric from the left (TOT respect the control CNT test). "Improv" (green to purple) means improvement of the TOT respect CNT; whereas "Degrad" (green to red) means degradation of the prediction. Points closed to each pilot site buoy, are in-situ buoys.**

Moderate wind-sea waves, though, are well captured in the NW Mediterranean. The J-22 storm had moderate wind-sea waves event at three pilot-sites. At GAL and TAR sites, the TOT run had better metrics of $H_{m0}$ and $H_{max}$ at the storm peak (21st - 22nd January 2022). At the same peak, TOT $H_{m0}$ improved close to 0.5 m (14%) at GAL-D and 0.25 m (16%) at TAR (**Figs. 14 and 15**); mainly due to the action of wind fields. Mean wave periods were low (close to 3s in TAR (**Fig. 15[b]**); and 5.5s in GAL (**Fig. 14[b]**)), because they were events driven by wind-sea waves, and the fetch associated to the wind direction was limited (around 640 km in GAL and 600 km in TAR).

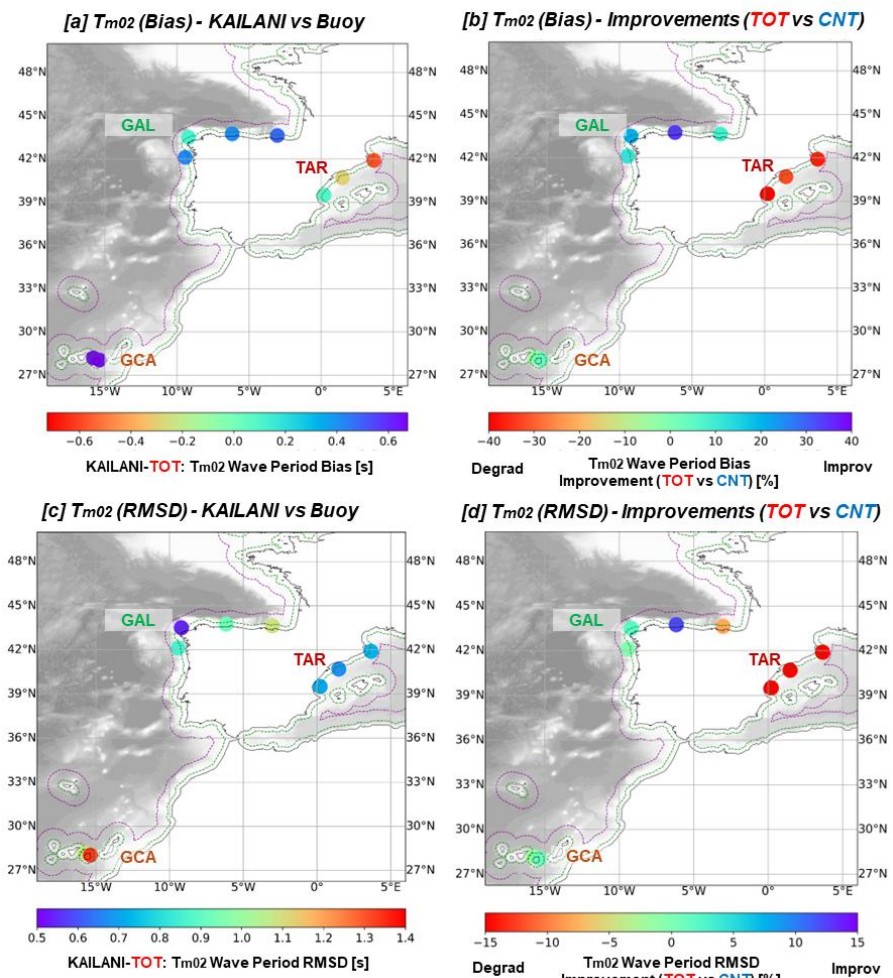

**Figure 11: Integrated $T_{m02}$ error metrics for the whole simulation period (2021/10 - 2022/01). Left panels [a,c] show the Bias and RMSD at the TOT experiment. Right panels [b,d] the % of improvement of the same metric from the left (TOT respect the control CNT test). "Improv" (green to purple) means improvement of the TOT respect CNT; whereas "Degrad" (green to red) means degradation of the prediction. Points closed to each pilot site buoy, are in-situ buoys.**

At Gran Canaria (GCA), the low-pressure center drove Eastern and SE winds (reaching 10-12 m/s), leading to SE waves reaching $H_{m0}$ of 1.8m at GCA (**Fig. 16**). Mean wave period remained short (close to 4s, **Fig. 16[b]**), as the fetch was limited from the Western Sahara coast to Gran Canaria (close to 200 km). Across the fetch, wind ANN predicted higher winds than IFS (note that in **Fig. 16[d]**). Hence, the $H_{m0}$ (**Fig. 16[a]**) and $H_{max}$ (**Fig. 16[c]**) in TOT and WND were better captured.

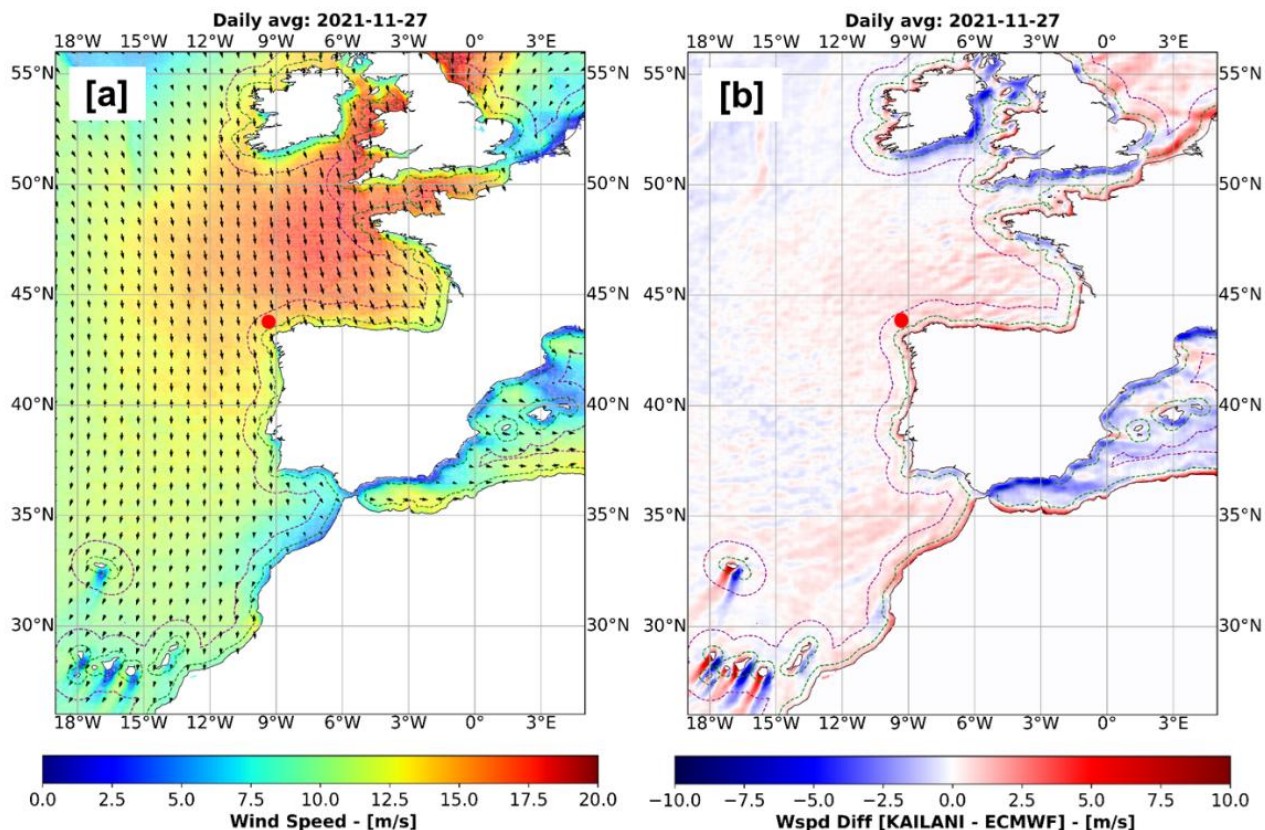

**Figure 12:** (Left) ANN-generated daily-averaged wind fields during the November 2021 event (storm peak, on 2021-11-27). The coloured areas the wind speed in m/s and the black arrows the wind vectors. (Right) Difference between KAILANI minus ECMWF-IFS daily-averaged wind speed. Units in m/s. White to red means that KAILANI predicts higher speeds than ECMWF-IFS; whilst white to blue refers that KAILANI predicts lower speeds. The red dot is the GAL-D buoy (see Fig. 13).

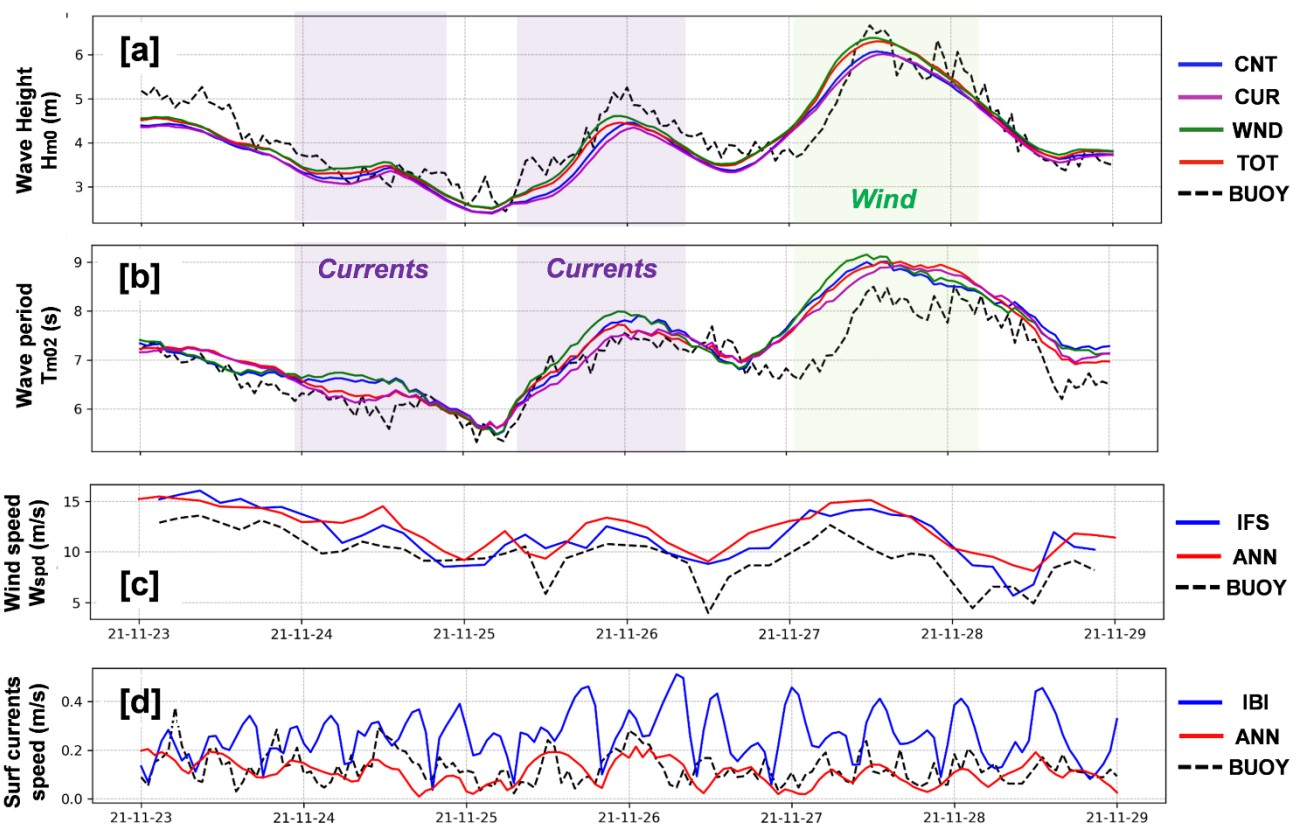

**Figure 13: Time series of the performance of the wave parameters and their associated forcings at GAL (Villano Buoy) under the November 2021 event. [a] $H_{m0}$ of the different experiments (see Table 2) vs the buoy. [b] Same as [a], but for the $T_{m02}$. The shaded time slices highlight good performance of the enhanced forcings, and the colour hints the main factor (be it wind [green] or currents [purple]). [c] Wind speed predicted by ECMWF-IFS (blue), the wind ANN (red) and the buoy measurements (dashed line). [d] Surface current speed predicted by IBI-PHY (blue), the currents ANN (red) and the buoy (dashed line).**

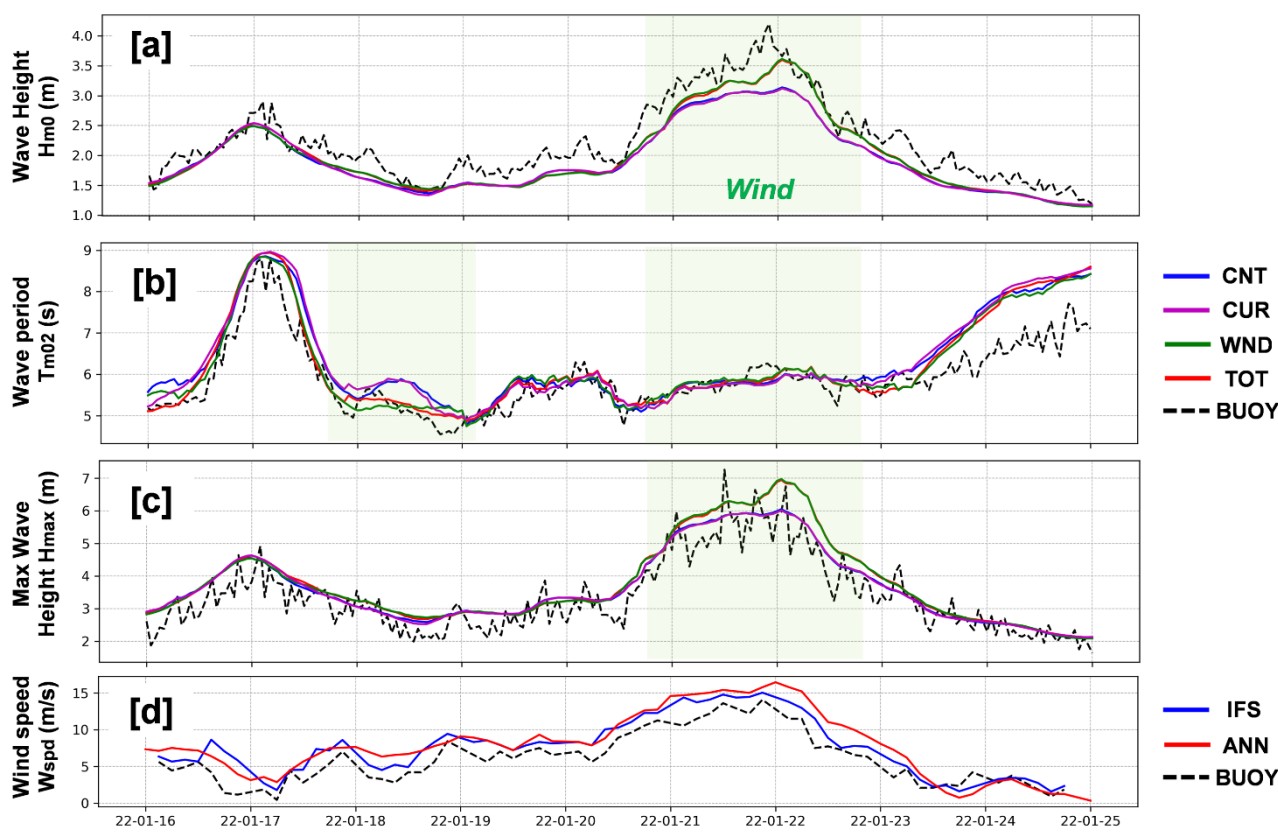

**Figure 14: Time series of the performance of the wave parameters and their associated forcings at Galicia (GAL) under the January 2022 event. [a] $H_{m0}$ of the different experiments (see Table 2) vs the buoy. [b] Same as [a], but for the $T_{m02}$. [c] Same as [a], but for the $H_{max}$. The shaded time slices highlight good performance of the enhanced forcings, and the colour hints the main factor (be it wind [green] or currents [purple]). [d] Wind speed predicted by ECMWF-IFS (blue), the wind ANN (red) and the buoy**
**measurements (dashed line).**

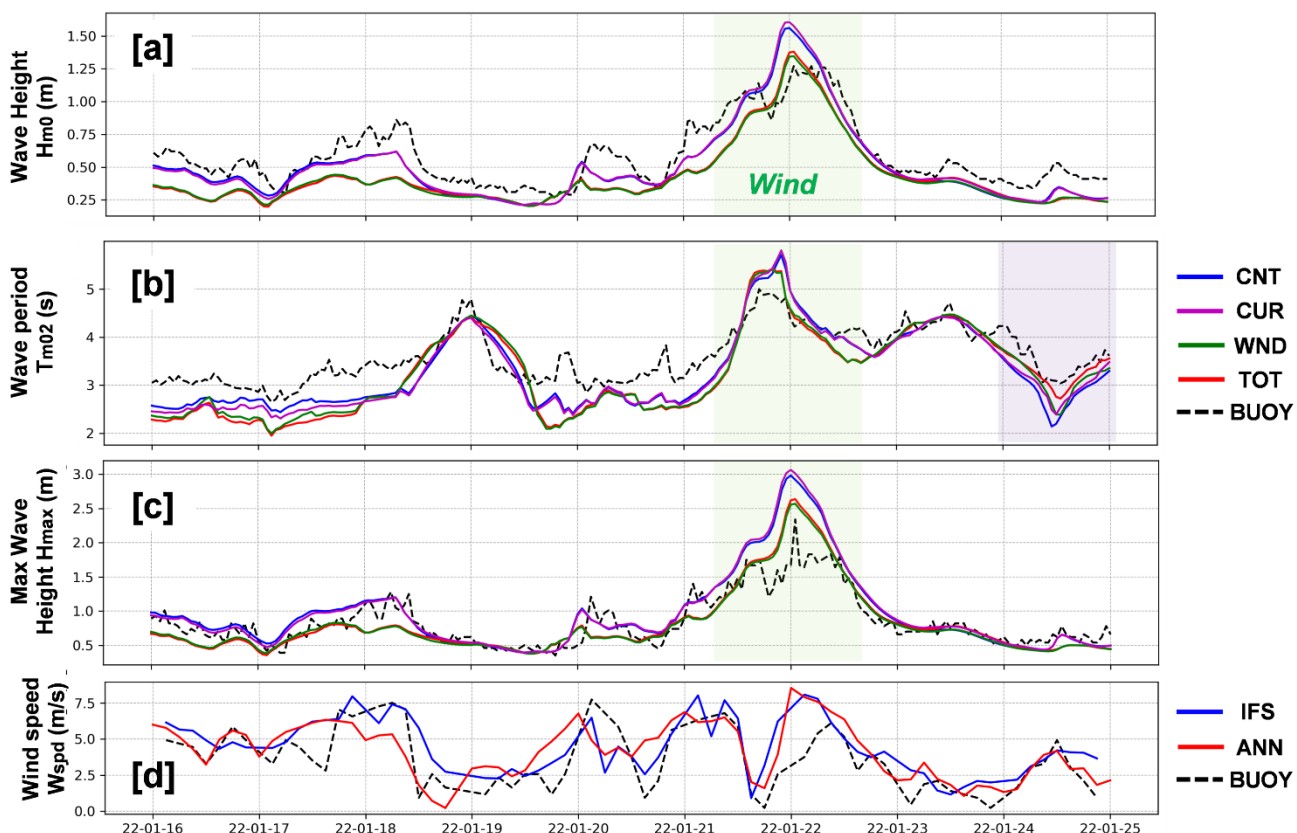

**Figure 15: Time series of the performance of the wave parameters and their associated forcings at Tarragona (TAR) under the January 2022 event. [a] $H_{m0}$ of the different experiments (see Table 2) vs the buoy. [b] Same as [a], but for the $T_{m02}$. [c] Same as [a], but for the $H_{max}$. The shaded time slices highlight good performance of the enhanced forcings, and the colour hints the main factor (be it wind [green] or currents [purple]). [d] Wind speed predicted by ECMWF-IFS (blue), the wind ANN (red) and the buoy measurements (dashed line).**

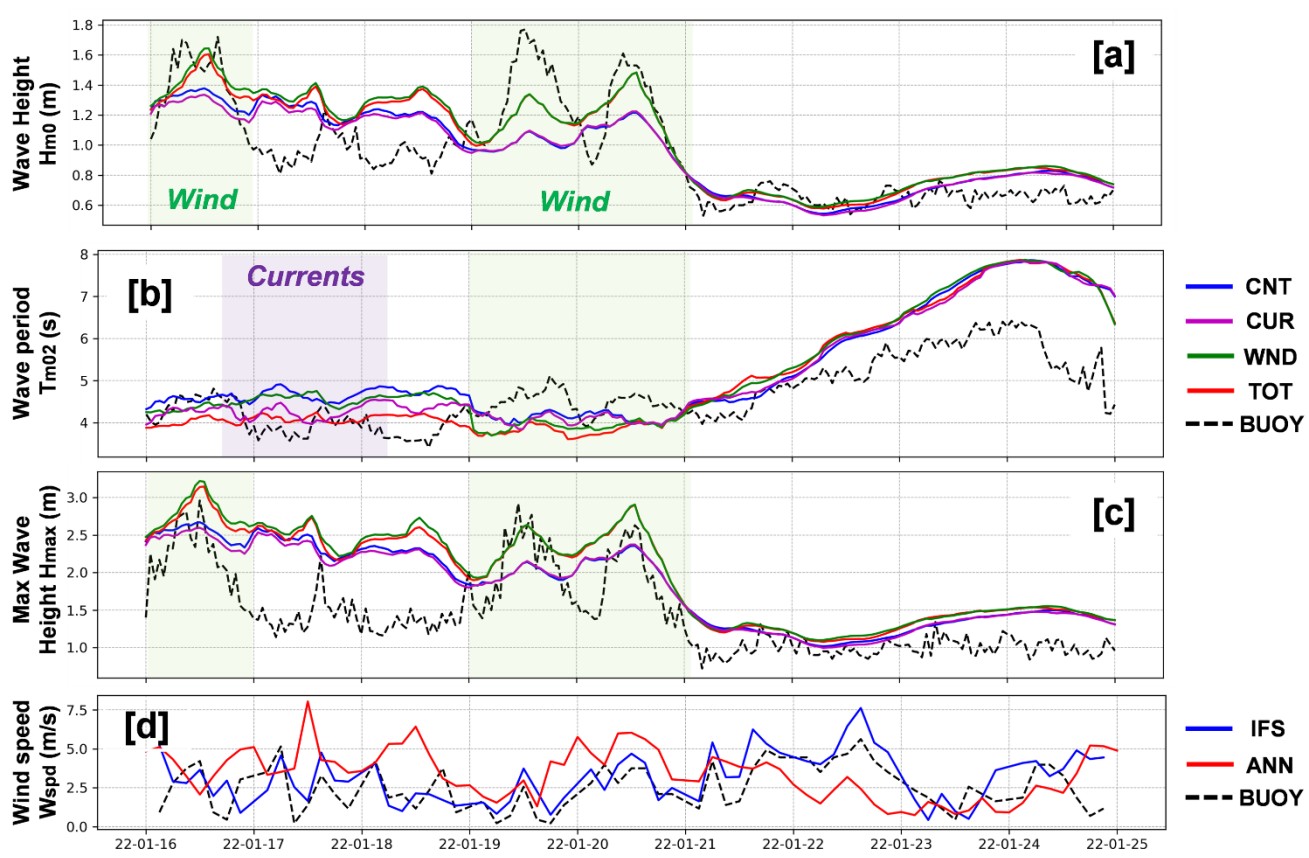

Figure 16: Time series of the performance of the wave parameters and their associated forcings at Gran Canaria (GCA) under the January 2022 event. [a] $H_{m0}$ of the different experiments (see Table 2) vs the buoy. [b] Same as [a], but for the $T_{m02}$. [c] Same as [a], but for the $H_{max}$. The shaded time slices highlight good performance of the enhanced forcings, and the colour hints the main factor (be it wind [green] or currents [purple]). [d] Wind speed predicted by ECMWF-IFS (blue), the wind ANN (red) and the buoy measurements (dashed line).

## 4. Discussion

The KAILANI methodology demonstrates improved coastal wave forecasts through enhanced ANN forcing, though its impact is limited in moderate weather conditions and conclusions are still preliminary during extreme events. Best performance was achieved using both ANN winds and surface currents, indicating that correction of both variables has positive impact. A key advantage of KAILANI is this ability to bound forecast uncertainty by minimizing forcing errors. While additional errors remain (as forcing biases are not fully mitigated) and other sources of uncertainty persist (e.g., in physical processes), the methodology offers valuable insights at the three pilot sites.

The wind ANN corrected wind fields across the entire computational domain, extending the impact of the methodology beyond the pilot sites to surrounding areas. For example, during storm Arwen, the wind ANN predicted stronger winds than

ECMWF-IFS over the Cantabrian Sea and Gulf of Biscay, resulting in more accurate modelling of the storm peak through improved wind-sea wave generation. A localized correction, just in Galicia, would have yielded a smaller improvement.

The Galicia and Cantabrian Sea regions benefited most significantly from the correction Among the three pilot sites, Galicia is the most representative in the training sample, including a share of the weather patterns at the region. At Gran Canaria and Tarragona, local effects, such as sheltering by the Canary Islands and mountain ranges near the Ebro valley, have a more prominent role.

In the NE Atlantic (Galicia and Gran Canaria), wind speeds were systematically overestimated in both the ECMWF-IFS and ESA SAR OCN L2 datasets (see **Subsection 3.1**). IFS vs. SAR biases were 1.35 vs. 0.96 m/s, and RMSDs were 1.93 vs. 1.49, respectively. Despite this, the wave model tends to underestimate wave heights, potentially due to air-sea transfer and wave generation parameterizations.

At the Tarragona pilot site, modelled wave heights are underestimated more substantially than in the operational IBI-WAV, which already tends to underestimate waves due to underestimated ECMWF-IFS winds in the NW Mediterranean. While SAR data showed good skill and coverage in moderate regimes, it tended to severely underestimate extremes, meaning the wind correction could still lead to wave height underestimation. Furthermore, the IBI-WAV wave generation parameterization is tuned for the Northeast Atlantic, not the Mediterranean.

As denoted in **Subsection 2.4**, a significant fraction of extreme events at Tarragona during the testing period were wind jets constrained by local topography (i.e. mountain ranges), for which the sample size was insufficient. While the wind ANN corrected the bias in the NW Mediterranean (from 0.89 to -0.07 m/s) with good skill under moderate winds, it underpredicted wind regimes above 10 m/s, exacerbating the IBI-WAV wave height underestimation. In the NE Atlantic, IFS and ANN wind biases were similar under moderate wind regimes.

This performance limitation stems primarily from the substantial cost of retrieving, processing, and preparing the Wind ANN dataset. Although SAR data from a wider area than the three pilot sites were included, it proved insufficient. While the training sample is representative of the Atlantic French, Iberian Peninsula, and Canary Island coasts, it is not adequately representative of the NW Mediterranean. Future work will focus on expanding the volume of Mediterranean SAR data or developing a dedicated model for the region.

The surface current ANN had a more limited impact on the wave solution, primarily affecting the Mean Wave Period (through frequency shifting) within the HFR coverage area. Outside this area, the impact was negligible. Furthermore, the distinct circulation patterns at each pilot-site presented a challenge for developing a unique generalized ANN model. Consequently, site-specific ANN models were implemented.

Total currents were used due to their gapless nature. While radial data could have been an alternative, it would have required additional methods for handling data spikes and gaps, requiring extra quality control procedures. Hence, it was prioritized a target dataset with fewer gaps and spikes, accepting a degree of smoothing inherent in total current data.

Although the ANNs significantly improved IBI-PHY performance (reducing biases in speed and direction), the scatter index remains relatively high (close to 50%, see **Subsection 3.2**). Considerable error persists during extreme events, precisely

when wave-current interaction processes are most influential on the wave solution. Future developments will explore expanding the training datasets and employing advanced strategies, such as deep generative models (see below), to enhance performance under data scarcity.

Several architectures were tested for the Wind and Surface current ANNs, but the proposed GAN and AE networks provided the best performance. For the Wind ANN, better results were achieved using a GAN than a CNN-based super-resolution network, such as a U-NET (Ronneberger et al., 2015). Although GANs are more complex, they partially alleviate the problem of defining a suitable loss function. Furthermore, the Wind ANN requires a Generative architecture because it not only increases the image resolution but also attempts to reduce persistent biases. Conversely, for the Surface Current ANN, the problem involves corrections in both space and time, making a GAN with Convolutional-LSTM layers difficult to train; therefore, an Autoencoder was selected.

Enhancing the KAILANI methodology involves more than expanding training datasets or adopting advanced ANN architectures. While SAR data is valuable for GAN training, its use is limited by the reliance on Geophysical Model Functions (GMFs) for wind retrieval. These GMFs, while computationally efficient (Portabella et al., 2002), are empirical relationships based on simplified air-sea interaction assumptions. Specifically, the GMFs used to derive wind speed and, critically, wind direction from SAR backscatter are subject to inaccuracies. These inaccuracies can be significantly exacerbated in complex coastal zones due to phenomena such as wave breaking, shallow water effects, and land contamination, which can alter the backscatter signal in ways not accounted for by the GMFs. For instance, in shallow waters, bottom topography can influence wave patterns, affecting the surface roughness and thus the SAR backscatter, leading to inaccurate wind retrievals (Lu et al., 2018).

A key drawback is the indirect nature of wind direction retrieval from SAR. Unlike wind speed, which is primarily related to backscatter intensity, wind direction is inferred from the azimuthal anisotropy of the backscatter, which is then related to wind direction through the GMF. This indirect retrieval is prone to ambiguities, particularly at low wind speeds or in areas with complex wind patterns, where the backscatter anisotropy may be weak or influenced by other factors like surface currents, rain cells, or biogenic slicks (Kudryavtsev et al., 2014, Melsheimer et al., 1998, Gade et al., 1998). This dependence on GMFs for both wind speed and direction introduces a source of uncertainty that propagates into the GAN training process, potentially limiting the achievable accuracy of the corrected wind fields. Furthermore, the limited temporal sampling of Sentinel-1 SAR (Torres et al., 2012), with its relatively low revisit frequency and tendency for acquisitions at similar local times, restricts the GAN ability to learn and correct temporal variations in the wind field. This sampling bias means the GAN is primarily trained to correct spatial inconsistencies between the ECMWF-IFS model and SAR-derived wind fields, with limited capacity to address temporal errors such as incorrect diurnal cycles or the timing of wind events. To mitigate some of these limitations, the integration of scatterometer data for offshore regions offers a potential solution. Scatterometers, such as those onboard MetOp satellites (ASCAT), provide direct measurements of wind direction along with wind speed, albeit at a coarser spatial resolution than SAR (e.g., typically 25-50 km for scatterometers compared to ~1 km for SAR) (Stoffelen, 1998). Combining scatterometer data with SAR-derived wind fields could provide a more

comprehensive training dataset for the GAN, allowing it to learn both high-resolution spatial patterns from SAR and broader-scale wind fields and direction from scatterometers. This combined approach could lead to a more robust correction of the ECMWF-IFS model, particularly in coastal areas where both fine-scale spatial details and accurate wind direction are crucial. Moreover, combining SAR data from different look directions could also help to reduce ambiguities in wind direction retrieval (Dagestad et al., 2012).

While ANN-enhanced forcings may exhibit improved skill, this improvement does not guarantee corresponding enhancements in wave forecasts across all scenarios. Operational wave models are typically calibrated for forcings with known, inherent errors and biases. Reducing some of these errors requires recalibration of model parameters to align with the new forcings (Tolman, 1998, Oladejo et al., 2025). Implementing KAILANI operationally would therefore imply (i) a comprehensive assessment of the new error characteristics and (ii) a recalibration of several physical processes, including wave generation and dissipation. This recalibration would not only improve integrated wave parameters but also provide more consistent wave spectra for downstream applications. Reduced errors in output spectra would positively impact coastal services, such as morphodynamic forecasting. In these applications, cumulative errors during extreme events can significantly degrade coastal predictions (Sánchez-Arcilla et al., 2014); for example, a 20% increase in wave error can lead to a nearly 50% increase in morphodynamic prediction error.

The increasing number of coastal downstream applications (El Serafy et al., 2023, Capet et al., 2020) highlights the need for accurate prediction of extreme winds, surface currents, and waves, a significant challenge that KAILANI aims to address. Extreme conditions face the persistent challenge of limited and potentially unrepresentative training data. Extremes are rare by nature; and this scarcity hinders the ability of standard ANNs to learn the complex, non-linear relationships between input and outputs. The impact on extreme events requires further assessment. To ensure consistent improvements, the methodology must be tested with a larger sample of storm events.

Deep generative models, such as normalizing flows (Kingma & Dhariwal, 2018) and diffusion models (Ho et al., 2020), are being investigated for their potential to generate synthetic extreme event data, augmenting the training dataset and improving the model's ability to generalize to unseen conditions. Furthermore, research on uncertainty quantification in deep learning is crucial for providing reliable confidence intervals for extreme event forecasts, allowing for more informed decision-making (e.g., Lakshminarayanan et al., 2017). Providing quantified uncertainty in short-term forecasts (Li et al., 2024) will allow more effective emergency planning, resource allocation for coastal protection measures, and timely evacuation orders during impending storm events (Kyrkou et al., 2022).

## 4. Conclusions

Wind and surface currents are key forcings of spectral wave models, directly influencing wave forecast accuracy. This contribution aimed to propose a methodology (termed KAILANI) for correcting these forcings. KAILANI feasibility has been tested as inputs for a regional wave forecasting system in the Iberian-Biscay-Ireland (IBI) area: Copernicus Marine IBI-

WAV NRT. Biases and errors in coastal winds and surface currents have been corrected with Artificial Neural Networks (ANNs) that were trained with Satellite Synthetic Aperture Radar (SAR) and High Frequency Radar (HFR) data. These ANNs can predict winds and currents at specific coastal locations that are blended with existing operational forecasts for winds (ECMWF-IFS) and surface circulation (IBI-PHY). Three pilot-sites have been selected at the IBI area due to its HFR data coverage and representation of the main features at European ocean waters: (i) Galicia, (ii) Gran Canaria Island (both at NE Atlantic) and (iii) Tarragona (NW Mediterranean Sea).

The wind ANN predicts surface winds by using a Generative Adversarial Network (GAN) that downscale ECMWF-IFS winds (1/8º) to the target dataset (S1A/S1B SAR OCN L2 products at 1/100º). Using SAR as reference, the ANN winds present RMSD reductions close to 35% respect to ECMWF-IFS, and improvements close to 3% for the scatter-index. Wind direction from the SAR product is inferred by ECMWF-IFS, so the Wind ANN shows not significant changes at this variable.

Subinertial surface currents are predicted with ANNs based on Autoencoders (AE) that uses hourly-averaged detided HF-Radar zonal and meridional data as target dataset; retrieved from Copernicus Marine In-Situ TAC. The input data includes (i) subinertial IBI-PHY currents, (ii) wind fields from ECMWF-IFS, (iii) Stokes-Drift and (iv) if available, HFR data from previous days. The AE improved the error metrics respect IBI-PHY at the three pilot sites, with similar accuracy: (i) speed and directional bias were close to 2 cm/s and 6º, respectively; (ii) speed RMSD was close to 11 cm/s and (iii) correlation close to 35% and 50%. The ANN exhibits better performance at extreme quantiles than IBI-PHY, but the scatter-index remains high (close to 50%).

September 2021 – January 2022 has been selected as the benchmark period, as IBI-WAV NRT underperformed and it also features weather regimes unseen by the ANNs. Winds and currents have been ANN-predicted and they have been blended with operational ECMWF-IFS and IBI-PHY forcings. IBI-WAV NRT has been run with the new forcings and a set of preoperational tests are being run to assess the feasibility of the ANN-generated forcings.

The ANN forcings have positive impact on the wave forecasts. Best performance has been found when using together Wind and Currents ANN forcings. Wind ANN had more impact than the Currents ANN; probably since wind fields are corrected along the whole IBI domain, whilst currents only at the pilot sites. Wind speed has an impact on the $H_{m0}$ and $T_{m02}$, but currents mainly have effect on $T_{m02}$. Wind ANN tends to decrease the overestimation of the ECMWF-IFS wind speed, leading to higher wave errors at the NW Mediterranean (e.g. the Wave Height ($H_{m0}$) bias increases close to 20%). Improving and extending the training dataset could help the wind model to generalize, thus alleviating the abovementioned issues. Integrated metrics throughout the benchmark period show that: (i) the $H_{m0}$ bias and RMSD improve around 10% and 5% at the NE Atlantic, respectively. (ii) At the same NE Atlantic, Mean Wave Period ($T_{m02}$) bias and RMSD improve 17% and 5%. While further assessment of the ANN forcings under extreme events is required, preliminary results indicate a potential for enhanced wave performance: during Storm Arwen at Galicia, $H_{m0}$ was corrected by approximately 0.5 m and $T_{m02}$ by around 0.4 s.

*Code and data availability*. Copernicus Marine IBI-WAV data is available at 10.48670/moi-00025, Copernicus Marine IBI-PHY data can be found at 10.48670/moi-00027. The in-situ observations can be retrieved from Copernicus Marine INS-TAC
(10.48670/moi-00036). Sentinel 1 OCN L2 data can be obtained at https://dataspace.copernicus.eu/.

*Author Contributions*. MGL led the paper, coordinated the KAILANI project and participated in the development of the surface current ANN model. JMGV was the main developer of the wind ANN and participated in the ANN for surface currents. LA participated in the ANN developments and led the IBI-WAV preoperational tests. AD and BG participated in
modelling the preoperational tests with IBI-WAV. JA participated in the development and application of the blending for surface currents. VA participated in the development of the Wind ANN model. SAC led and participated in the product-quality assessment of the simulations. RA and MS participated in the ANN developments and the uptake of the KAILANI methodology for the IBI-WAV service. All authors provided comments and corrections to the initial version of the paper.

*Competing interests*. The authors declare no competing interests.

*Acknowledgements*. This work has been carried out as part of the Copernicus Marine Service KAILANI project (contract no. 21036L04B-COP-INNO SCI-9000). Copernicus Marine Service is implemented by Mercator Ocean in the framework of a delegation agreement with the European Union. This study has been conducted using E.U. Copernicus Marine Service
Information (10.48670/moi-00036, 10.48670/moi-00027, 10.48670/moi-00025). The authors would like to thank the ESA for the Sentinel-1 SAR Level 2 products and the ECMWF for the ECMWF-IFS forecasts.

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
