# Peer review of "Enhancing coastal winds and surface ocean currents with deep learning for short-term wave forecasting"

_EGUsphere, 2025_

## Author Comment (AC1)

**Egusphere-2025-657 - Answer to reviewers**

**REPORT#1**

Review of "Enhancing coastal winds and surface ocean currents with deep learning for short-term wave forecasting", manuscript egusphere-2025-657

This manuscript presents a practical approach to improving the performance of numerical wave models by correcting their forcing fields—namely wind and surface currents—using Artificial Neural Networks trained on remote sensing data such as SAR and HFR. The methodology is applied and validated at multiple pilot sites, demonstrating consistent and significant improvements across several key metrics. The corrected forcings lead to better wave height and period predictions, both under normal and extreme conditions. Overall, this work is methodologically sound, relevant to the field of operational ocean forecasting, and contributes meaningful advancements in the integration of remote sensing with data-driven modeling techniques. Therefore, after making some appropriate revisions (mainly formatting issues), I believe this manuscript is suitable for publication. Here are some of my comments about the manuscript.

We thank Reviewer #1 for the positive feedback and for taking the time to review our manuscript. We have tried to address all your suggestions in this revised version, and we believe they have significantly improved the manuscript. We hope this revision meets with your approval.

**Major comments:**

1. I noticed that a GAN-based architecture was used for wind field correction, while an autoencoder-like structure was adopted for surface current correction. Could the authors comment on the rationale behind selecting these different architectures for the two tasks? Also, were other model types explored or compared during the development process, e.g., if a CNN-based super-resolution network was used directly instead of a GAN model (i.e., SAR is used directly as a target, with the network output calculating an RMSE-like loss), would this be any less effective? (Note that there is no need for the authors to add additional experiments here, just a brief discussion)

We thank the reviewer for this comment regarding our architectural choices. We agree that the rationale behind selecting different network types for the ANN wind and current corrections requires a clear explanation. We have incorporated the following text into the Discussion section:

Several architectures were tested for the Wind and Surface current ANNs, but the proposed GAN and AE networks provided the best performance. For the Wind ANN, better results were achieved using a GAN than a CNN-based super-resolution network, such as a U-NET (Ronneberger et al., 2015). Although GANs are more complex, they partially alleviate the problem of defining a suitable loss function. Furthermore, the Wind ANN requires a Generative architecture because it not only increases the image resolution but also attempts to reduce persistent biases. Conversely, for the Surface Current ANN, the problem involves corrections in both space and time, making a GAN with Convolutional-LSTM layers difficult to train; therefore, an Autoencoder was selected.

2. Figures 2 and 3 clearly illustrate the model architectures, and they are generally well-presented. However, the diagrams could be further improved by including more detailed information on the data dimensionality. For instance, adding the input and output shapes at the beginning and end of each model—either directly in the figures or in the accompanying text—would help readers better understand how the data is transformed through the network. This additional context would make the architecture more transparent and informative, especially for those interested in replicating or adapting the models.

We thank the reviewer for this constructive suggestion. We agree that providing clear information on data dimensionality is standard practice and greatly enhances the reproducibility and clarity of the figures.

We have addressed this by adding further explanations regarding the input and output shapes in the manuscript (specifically in Subsections 2.2 and 2.3).

Regarding the direct inclusion of tensor sizes in Figures 2 and 3, we opted to keep the diagrams schematic for the following reasons:

• Surface Currents ANN: Each pilot site model architecture uses different tensor sizes for input and output because these dimensions depend on the unique spatial extent of each High-Frequency Radar (HFR) domain. Since the tensor size varies across the three pilot areas (e.g., the HFR at Galicia has a larger extent than the one at Gran

- Canaria), including these variable values directly in a single figure could easily lead to confusion for the reader. Therefore, for the sake of clarity, we chose to detail these site-specific dimensions within the accompanying text instead.
- ANN Wind: This model involves a super-resolution task where the input images are a fixed 10×10 pixels, and the output is a fixed 100×100 pixels.
- 3. I suggest the authors include a brief subsection (such as **2.6 Error Metrics**) in Section 2 that summarizes all the error metrics used throughout the manuscript. This summary should provide the definitions and explicit formulas for each metric (e.g., RMSD, bias, correlation, etc.). Doing so would enhance clarity and help readers better understand the evaluation criteria, especially those who may not be familiar with all the statistical indicators applied.

We thank the reviewer#1 for this suggestion. We have included a new subsection 2.6 Error Metrics.

**Detailed comments:**

1. The authors should pay close attention to citation formatting throughout the manuscript. For instance, in Line 70, the citation "(Gurgel et al., 1999)" is correctly formatted, but in other places (e.g., Line 52: "Hauser et al. 2023"), the comma after "et al." is missing. Such inconsistencies should be carefully checked and corrected. Additionally, figure references should follow the format of the journal—"Figure X" is appropriate at the beginning of a sentence, while "Fig. X" should be used elsewhere. Some citations are also inconsistently bolded, which should be standardized to maintain uniform formatting. Issues like this hopefully the authors can address them in a revised manuscript

We thank the reviewer#1 for this remark. We have revised the whole manuscript and we have standardized citation formatting and Figure references.

2. In Line 188, the authors refer to "training/validation datasets" in the context of evaluating model performance. However, if the dataset mentioned here is used solely for post-training evaluation rather than during model training for purposes like early stopping or hyperparameter tuning, it would be more accurate to refer to it as a "test dataset" rather than a "validation dataset". Similarly, the term "validation period" used later in the manuscript should be revised to "test period" or "evaluation period" which may help avoid confusion.

We thank the reviewer#1 for this remark. We have revised the entire manuscript and changed all references to "validation dataset" to "testing dataset."

3. In Lines 301–314, multiple date formats are used inconsistently, such as "January 2021 – January 2023", "Sep 2021 – Jan 2022", "25th – 27th November", "November 26–27, 2021", and "20th–23rd January 2022". I recommend standardizing the date format throughout the manuscript for consistency and improved readability.

We thank the reviewer#1 for this remark. We have standardized the date format throughout the manuscript with this convention: 20th January 2022.

4. Many formatting inconsistencies can be noted in Fig. 5 and Fig. 9. For example, for the scatterplot, while the scale intervals are numerically the same, the x-axis has a sparser scale density than the y-axis. Also, the gridlines are either present or absent. The unit notation is also different between the two: one uses '[m/s]' while the other uses '(m/s)'. In addition, the 1:1 reference line is drawn in different colors - red on the left and green on the right - which may cause unnecessary distractions. Standardizing these visual elements will enhance the overall coherence and presentation quality of the charts.

We agree with Reviewer #1 that Figs. 5 and 9 could have been improved. Thus, we have redone them, following Reviewer #1 recommendations.

5. In Fig. 7, which displays both positive and negative deviations, I suggest adjusting the color bar so that the central (white) point is explicitly labeled as 0. Additionally, using symmetric tick values for positive and negative ranges—ideally with a limited number of decimal places (e.g., [..., -7.6, -3.8, 0.0, 3.8, 7.6, ...])—would improve both the readability and the aesthetic quality of the figure. Also, the word spacing in the subheading of this image is odd.

We agree with Reviewer#1 that Fig. 7 required improvement. Thus, we have redone it, following Reviewer #1 recommendations. We have replaced the colour scale by a symmetric one.

6. In 13–16, there is a noticeable mismatch in the color tone between the plot lines and their corresponding legend entries—for example, while both may be shades of blue, one appears significantly lighter or darker than the other. If the legends were added during figure post-processing, using a color picker tool to precisely match the tones would improve the visual coherence. Although this does not affect the scientific interpretation, ensuring consistency in color tones would enhance the professionalism and clarity of the figures.

We thank Reviewer #1 for this remark. We have revised Figures 13 through 16 to correct the mismatch between the plot lines and their corresponding legend entries.

7. Throughout the manuscript, there are noticeable inconsistencies in figure formatting that should be addressed. For example, multiple styles are used for subfigure labels, including (1), [1], and (i), which creates confusion and detracts from the overall professionalism. Additionally, figure and table titles vary in formatting—some are in italics while others are in regular font, which should be standardized. Moreover, the resolution of several figures appears to be quite low, with visibly pixelated text and labels.

We thank Reviewer #1 for this remark. We have tried to address the inconsistencies in figure formatting. We have also tried to increase the quality of those figures with pixelated text and labels.

---

## Author Comment (AC2)

**Egusphere-2025-657 - Answer to reviewers**

**REPORT#2**

This paper describes a method for adjusting operational surface fields, specifically wind speed and current speed, to minimise forecast errors in the coastal zone. Artificial neural networks trained with radar data from land stations (HFR) and satellites (SAR) adjust forcing fields. Overall, the paper reads well and contains all relevant references. However, the vast amount of acronyms limits readability to some extent. The methodology is reasonably well documented, and the verification of surface winds and currents appears sound. I appreciate the attempt to separate errors in the forcings that drive the wave model.

We thank Reviewer #2 for the positive feedback and for taking the time to review our manuscript. We have tried to address all your suggestions in this revised version, and we believe they have significantly improved the manuscript. We hope this revision meets with your approval.

Verification for "short-term wave forecasting" is missing. Perhaps this is out of scope, then the title would need to be edited accordingly. Wave verification is too brief and could benefit from a more detailed analysis, other than a case study. Time series plots presented in Figure 13-16 did not convince me of an improved forecast system. Numerical weather prediction models feature some variability in skill over a complete seasonal cycle. Training and verification for a particular season can introduce seasonal bias in the correction method.

We thank Reviewer #2 for this important remark. Our aim was to focus on the Neural Networks for correcting wind and currents, and we acknowledge that a full verification of short-term wave forecasting is indeed beyond the scope of this study.

Our testing with the IBI-WAV model was primarily intended to demonstrate the overall potential and feasibility of the methodology. We consider that a proper, comprehensive validation of the forecast skill warrants dedicated attention and a specific follow-up study.

We are aware that training and verifying data from a particular season can introduce seasonal biases. To mitigate this, we intentionally selected a verification period that was as out-of-sample as possible, which partially explains the modest relative performance of the methodology shown in the figures.

We agree with the reviewer that two isolated extreme events are not fully representative of the system's performance under all extreme conditions. This is why we have highlighted throughout the revised text that any results and conclusions regarding extreme events should be considered as preliminary.

We are currently working on a follow-up study to test the feasibility of this methodology using more comprehensive benchmarks, including a more detailed analysis of extreme event data.

The introduction could benefit from a paragraph on the expected impact on the wave model skill, not just for the wind generation part, but also for the effect of currents on wave predictability.

We thank the Reviewer#2 for this remark. We have added the following text in the Introduction:

Another main forcing at the coastal zone are the surface currents. Including currents in spectral wave models can reduce the errors on significant wave heights by more than 30% in some macrotidal environments, such as at the coast of Brittany (France) (Ardhuin et al., 2012). Wave-current interaction can affect (i) refraction due to currents, (ii) shoaling, and (iii) current-driven frequency shifting (Staneva et al., 2017; Cavalieri et al., 2018; Law Chune and Aouf, 2018; Bruciaferri et al., 2021; Calvino et al., 2022). While numerical wave models have demonstrated considerable skill in predicting wave conditions, the predictive skill of coastal circulation models remains comparatively lower, particularly in complex coastal regions (Fringer et al., 2019, García-León et al., 2022).

**Specific comments**

Referencing in text uses inconsistent punctuation and contains et al. year, et al., year and author, year.

We thank the Reviewer#2 for this remark. We have revised the whole manuscript and we have standardized citation formatting and Figure references.

Line 29-30: "Third generation spectral wave models ..., as they address wave generation and propagation". Third-generation models explicitly represent the nonlinear wave-wave interactions. Generation and propagation refer to second-generation models.

We thank the Reviewer#2 for this remark. Please find the revised version of the text:

Third-generation spectral wave models are suitable for the regional scale because they address wave generation, propagation, and non-linear interactions (WISE group, 2007).

Line 31-32: "three factors". What about the representation error? In the coastal zone, many processes are often overlooked, such as wetting and drying, river discharge, variable water depth, changes in beach profile, and seagrass, among others.

We thank the Reviewer #2 for this remark. Please find the revised version of the text:

They are computationally affordable and reliable for many applications (Capet et al., 2020). However, errors tend to increase at the coast, due to the compound effect of many factors: (i) accuracy in the forcings; (ii) limitations in the physical parameterizations; (iii) overlooked physical processes (e.g. river discharge, coastal morphodynamics, wave-biota interaction, wet-and-dry), (iv) model discretization and numerical schemes, etc.

Line 34: "(ii) reduce biases and errors in the inputs (Durrant et al 2013) ...". Complementary to Durrant et al., Zieger (2025) assessed forecast errors in an operational wave model. (https://doi.org/10.1071/ES25010).

We thank the Reviewer #2 for this reference. We have added Zieger (2025) in the Introduction.

Line 38-39: "Reducing biases and errors in the forcings have certain advantages...it does not require changing the physics." The parameterisations that represent the physics will naturally contain some form of bias that stems from model development. The underlying tuning parameters are bound to the frequency of model fields (daily, 6 hours, 3 hours, hourly) and the underlying statistic (i.e., cumulative, average, instantaneous). As a result, one has to change the physics kind of.

We thank the Reviewer #2 for this remark. Please find the revised version of the text:

Reducing biases and errors in the forcings has certain advantages over other strategies, because it does not increase computational time by changing the model resolution. Lower forcing errors also bounds a source of uncertainty, assisting in the implementation of the other two strategies. Main inputs in the spectral wave models are (i) wind fields, (ii) surface ocean currents and (iii) bathymetry. At regional scale models, the bathymetry is considered as a static forcing, because the rate of change for short-term forecasting (i.e. a few days) is only significant at shallow waters. Hence, for the forecast window considered, winds and currents can be considered as the main dynamic forcings.

Line 84: "spectacular growth" does not sound well. Perhaps "exponential growth".

We thank the reviewer#2 for this remark. We have added "exponential growth" in the revised version.

Line 90: "concrete". This word does not feel right and does not add clarity to the sentence (omit). There is a slight overuse of the word 'concrete' in the manuscript.

We thank the Reviewer#2 for this remark. All mentions to "concrete" have been removed.

Line 128: Could you please provide more specific details on the ST4 configuration? Could you state the value for BETAMAX used?

We thank the Reviewer#2 for this remark. Please find the revised version of the text:

Wave physics within IBI-WAV are parameterized using the ST4 formulation (Ardhuin et al., 2010), accounting for energy dissipation due to wave breaking and swell decay. The MFWAM implementation of ST4 is further enhanced by incorporating a Phillips tail spectrum to accurately represent the high-frequency portion of the wave spectrum. The  $\beta_{max}$  tuning parameter, that adjusts the transfer of energy and momentum from the wind to the waves, is 1.48. This is consistent with other applications in the literature (i.e. 1.39 in Valiente et al., 2023; 1.52 in Ardhuin et al., 2010 or 1.75 in Alday et al., 2021).

Line 186-187: "It has been discarded images" is gibberish. Please rephrase the sentence.

We thank the Reviewer#2 for this remark. We have rewritten this sentence.

Line 360-361: "So, it can be concluded that the ANN is able to predict winds that are closer to the SAR data than IFS." Would one not verify against an independent dataset (also Figure 5)? The ANN training would be well aware of the characteristics and structure of SAR data.

We thank the Reviewer #2 for this remark. Our main aim with the Wind ANN was to show that the GAN can effectively mimic the SAR data. The SAR data, when compared with an independent data source (in-situ buoy), presented better model skill (please refer to Figure 1). Note also the data scarcity in the area (575 valid time points in almost 5 years). This is one of the reasons that we required to keep SAR data from the Mediterranean and the NE Atlantic within the same training dataset. Another reason for this choice of training dataset was to obtain a homogeneous response and avoid disparate wind predictions across different basins. In current research, we are trying to add more SAR data, but further assessment is needed.

**Figure 1.** (up) Results of comparing SAR data with the in-situ buoy, and (low) comparing ECMWF-IFS with in-situ buoy. Note that the Wind ANN mimics well the SAR data, and the error metrics of the SAR are lower than ECMWF-IFS.

Line 479: "Benefits from the ANN forcing are more remarkable during extreme events. During storm Arwen, strong Northern winds at the Northern Iberian Peninsula (Figure 12a) ..." What is the rationale for showing daily-average marine wind speed to depict extreme events? The most pronounced feature in wind speed difference (Fig. 12b) is the north-west south-east shift around coastlines. In addition, wind speed verification at the GAL station (Fig. 13 and Fig. 14) indicates that surface wind speed is consistently overpredicted. At the same time, the responding wave field exhibits lower magnitudes for the duration of the storms. On the Mediterranean side, GCA station, is not able to capture the double peak on 19th and 20th January. Can you elaborate on this?

We thank the Reviewer#2 for this remark. We have added the following text in the revised version:

Overall improvement is also suggested during the two extreme events. However, because the sample of storms is limited, the results and interpretations must be considered as preliminary. **Figure 12** illustrates the expected order of magnitude of the differences between

the ECMWF-IFS and the Wind ANN products. These daily-averaged differences are representative of how the Wind ANN corrected the wind forcings during the peak of Storm Arwen. Furthermore, they provide qualitative insight into the reasons for the over- or underperformance observed in the wave results for both the TOT and WND experiments. During storm Arwen, strong Northern winds at the Northern Iberian Peninsula (Fig. 12[a]) generated Northern wind-sea waves, that were reinforced with NW swell waves. The integrated  $H_{m0}$  of this mixed sea state can be seen in Fig. 13. The storm peak was better handled (Fig. 13[a]), because at the wind-sea generation area, the ANN Winds predicted more intensity than ECMWF-IFS. The wind speed time series at GAL-D does not show much difference (Fig. 13[c]), but Fig. 12[b] shows clearly the extra wind energy at the area (red marked region), that lead the TOT and WND simulations to capture better the storm peak at  $27^{th}$  November 2021. Note also that the NW Mediterranean has less energy in KAILANI than in IFS (i.e. the blue area in Fig. 12[b]), suggesting the same issues mentioned in the general performance.

The Generative Adversarial Network (GAN) corrects wind speed but does not generate atmospheric patterns that strongly differ from ECMWF-IFS, leading to similar spatial structures.

We also note the specific case of the Gran Canaria (GCA) station. In this area, even the control simulation (ECMWF-IFS winds) was indeed unable to reproduce the observed moderate waves. The presence of the double peak at GCA is an interesting case. While the GAN successfully corrects systematic biases, it currently appears unable to reproduce highly localized wind effects. Since the waves were primarily wind-sea, the discrepancy could stem either from the local winds not being properly modeled or from the spectral wave model not accurately handling wave growth. As the reviewer correctly pointed out, the ANN's enhancement seemed to be a minor improvement that primarily involved increasing the wind speed in that zone.

However, any enhancements achieved under extreme events should be considered preliminary, and further research is needed for proper validation. This statement has been added throughout the text.

Line 671: "The ANN forcings have positive impact on the wave forecast, especially under extreme events." To be honest, the results for significant wave height seem to reduce the variability in the signal. Peaks are less pronounced in Figure 15, and almost no signal in the time series in Figure 16. In general, a diurnal signal appears to be present in the observed wave field, which is lost in the simulation.

We thank the Reviewer#2 for this remark. We agree that the variability is not the same as in the in-situ buoy, but this low variability is also present in the control simulation.

**This statement has been removed in the revised version.**

Line 674: "Wind ANN tends to decrease the overestimation of ECMWF-IFS wind speed". Is this correct? Time series plots in Figure 13-16 indicate that ANN winds are frequently higher than ECMWF and observations.

We thank the Reviewer#2 for this remark. This statement is correct regarding the Tarragona site, where the predicted ANN winds tend to be lower. Specifically, Figure 15 shows that the ANN winds are lower at the buoy location. Additionally, Figure 12 illustrates that the NW Mediterranean is an area where ECMWF has consistently higher values than IFS (indicated by the blue zones). The reasons are the same as those noted in the Line 360-361 specific comment: SAR data has better agreement than the IFS but exhibits lower overestimation in the NW Med. Furthermore, as was also mentioned, the IBI-WAV parameters are tuned for the NE Atlantic and therefore do not perform properly in the NW Mediterranean.